# Engineering glycosyltransferases into glycan binding proteins using a mammalian surface display platform

Ryoma Hombu[1], Lauren E. Beatty[2], John Tomaszewski[3], Sheldon Park[1] & Sriram Neelamegham [1,2,4] ✉

Traditional lectins exhibit broad binding specificity for cell-surface carbohydrates, and generating anti-glycan antibodies is challenging due to low immunogenicity. Nevertheless, it is necessary to develop glycan binding proteins for single-cell glycosylation pathway analysis. Here, we test the hypothesis that protein engineering of mammalian glycosyltransferases can yield glycan-binding proteins with defined specificity. Introducing an H302A mutation, based on rational design, into porcine ST3Gal1 abolishes its enzymatic activity, but results in a lectin that specifically binds sialylated core-2 O-linked glycans (Neu5Acα2-3Galβ1-3[GlcNAc(β1-6)]GalNAcα). To improve binding, we develop a mammalian cell-surface display platform to screen variants. One ST3Gal1 mutant (sCore2) with three mutations, H302A/A312I/F313S exhibits enhanced binding specificity. Spectral flow cytometry and tissue microarray analysis using sCore2 reveal distinct cell- and tissue-specific sialyl core-2 staining patterns in human blood cells and paraffin-embedded tissue sections. Overall, glycosyltransferases can be engineered to generate specific glycan binding proteins, suggesting that a similar approach may be extended to other glycoenzymes.

Glycosylation is a ubiquitous molecular modification that regulates diverse biological processes including molecular recognition, cell adhesion, and signaling[1,2]. Glycan structure changes also serve as biomarkers of cell differentiation and disease[3,4]. Due to these basic science and translational roles, the measurement of cell surface glycans has broad biomedical importance. Such measurements are, however, complicated due to the stereochemistry and branched nature of glycans. While mass spectrometry is well suited to identify glycan compositions and topologies particularly for abundant carbohydrate entities, detailed structural analysis and linkage specification is challenging and low abundant entities are missed. Complementary molecular tools that recognize specific glycan epitopes or sub-structures associated with biological transformation are thus needed.

Lectins derived from prokaryotes and eukaryotes have been used for glycan recognition in biochemical assays[5,6]. These glycan-binding proteins (GBPs), however, often lack binding specificity in that they commonly bind many related terminal epitopes, and currently available lectins do not cover all known glycan epitopes. Carbohydrate-binding modules (CBMs) are well-known GBPs that are part of glycan-processing enzymes that bring enzymes and substrates together to promote reactions. However, the CBMs usually recognize glyco-homopolymers or terminal monosaccharides, and they lack the ability to bind more complex structures[7,8]. Similarly, lectins derived by engineering glycosidases also only bind terminal antigens[9,10]. Anti-carbohydrate antibodies have been developed as an alternative, but the low immunogenicity of carbohydrate antigens

[1]Department of Chemical and Biological Engineering, University at Buffalo, State University of New York, Buffalo, NY, USA. [2]Department of Biomedical Engineering, University at Buffalo, State University of New York, Buffalo, NY, USA. [3]Department of Pathology and Anatomical Sciences, University at Buffalo, State University of New York, Buffalo, NY, USA. [4]Department of Medicine, University at Buffalo, State University of New York, Buffalo, NY, USA. ✉e-mail: neel@buffalo.edu

makes it challenging to derive monoclonal antibodies (mAbs) against arbitrary glycan epitopes[11,12]. Protein engineering approaches for GBPs have been attempted to engineer nature-derived GBPs to increase affinity or to convert specificity to related glycan epitopes[13,14]. However, these attempts have suffered from promiscuous binding to other glycan structures and reduced affinity. The shallow interaction interfaces of these nature-derived GBPs can also promote rapid dissociation of bound glycans due to competition with other molecules such as water. Finally, bacterial adhesins containing Siglec-like domains have been prepared as sialoglycan-binding lectins, and these broadly recognize Neu5Ac(α2-3)Gal and related epitopes[15,16].

Besides naturally occurring lectins, attempts have been made to endow other generic proteins with glycan-binding properties. For example, the DNA-binding protein, Sso7d was engineered through directed evolution to bind Thomsen-Fridenreich (TF) antigen. However, the engineered protein exhibited cross reactivity with unknown ligands[17]. Type B lamprey variable lymphocyte receptors (VLRBs), lambodies, were also engineered with some success to develop GBPs, but the ability of the leucine-rich zipper to bind arbitrary human glycan structures remains unknown[18,19], particularly as these are products of broad screening efforts rather than rational, predictive design. Overall, as the binding pockets of generic proteins are not naturally optimized for carbohydrate recognition, engineering them as GBPs can result in low affinity binders. Additional scaffolds with deep and extended glycan binding pockets are needed. Strategies to develop synthetic lectins in a predictive manner, rather than screening, would also be advantageous.

To address the above limitations, we test the hypothesis that glycosyltransferases (GTs) can be engineered to yield GBPs. In this regard, the binding pocket of mammalian GTs is evolutionarily optimized to accept specific glycans in order to synthesize a range of mammalian $N$-glycans, $O$-glycans, and glycolipids[20]. While pseudoenzymes lacking enzyme activity can recognize carbohydrate epitopes[6], focused efforts to engineer glycosyltransferase for this purpose are absent. We hypothesize that the modification of the GT interface may result in loss of enzyme activity but may favor either substrate or product binding. To test this, we engineer porcine ST3Gal1 (pST3Gal1, β-galactoside α2,3-sialyltransferase 1), a member of the GT29 CAZy family whose crystal structure is known[21]. We choose pST3Gal1 as it has a large binding pocket that can simultaneously accommodate the donor CMP-Neu5Ac (Cytidine-5-monophospho-$N$-acetylneuraminic acid) and TF-antigen disaccharide acceptor (Galβ1-3GalNAcα). Consistent with our proposition, we observe that the introduction of an H302A mutation into pST3Gal1 through rational design leads to a lectin that specifically binds sialyl core-2 $O$-glycans. We develop a new mammalian surface display platform to screen for additional variants. This results in a mutant H302A/A312I/F313S (sCore2 lectin) that displays even stronger binding for the sialyl core-2 epitope. Compared to traditional GBPs, this lectin displays unique binding patterns to different human peripheral blood cell types, normal human tissue and tumor sections. Overall, the study establishes a rational design and high-throughput screening strategy to convert GTs into GBPs. This strategy may be applied to other glycosyltransferases from diverse species.

## Results

### Conversion of Fc-pST3Gal1 (PS1) into the glycan-binding H302A

Upon comparing the molecular recognition surface area of pST3Gal1 with respect to other GBPs and glycosidases, it is apparent that the pST3Gal1 ligand-binding interfacial area (~540 Å²) is larger compared to the other entities (283–502 Å²) (Supplementary Fig. 1). This is consistent with the notion that lectins and glycosidases mostly only bind terminal residues and they exhibit broad binding specificity, whereas GTs display a more intricate binding preference.

We tested the hypothesis that the large GT binding interface may afford opportunities to generate lectins with novel binding properties that recognize the natural reaction substrate or product of the parent enzyme (Fig. 1). To test this, we created a set of three Fc-fusion proteins with the GT catalytic domain expressed at the C-terminus to mimic their presentation in the Golgi (Fig. 1a, and Supplementary Fig. 2a): i) The truncated pST3Gal1-Δ59 wild-type enzyme lacking N-terminal cytoplasmic and transmembrane domains (abbreviated PS1)[22]. ii) H302A, a PS1 variant that we hypothesized would be inactive due to loss of highly conserved H302 in motif 3. We posit that this mutant would, however, retain binding capacity for acceptor substrate or product as the mutation targets the hydrogen bond formed with the phosphate group of CMP-Neu5Ac[23], iii) Dead (Q108A/Y233A/Y269F), a non-binding mutant which cannot bind its TF antigen acceptor, Gal(β1-3)GalNAcα, due to loss of essential hydrogen bonds and π-π interactions (Fig. 1b)[24]. All proteins were expressed in dimeric form with a flexible Gly-Ser hinge in order to mimic natural lectin oligomerization[25–27] using a lentiviral vector containing a secretory Von Willebrand factor signal peptide, a 6xhis-tag for purification, and an enterokinase cleavage site. These proteins were expressed along with Fc-CBM40, a Fc-fusion protein containing a sialic acid-binding bacterial carbohydrate-binding module, and its tandem variant Fc-diCBM40 (Supplementary Fig. 2b, c)[28]. PS1, H302A, Dead, and Fc-CBM40 expressed well upon transient transfection into HEK293T cells at ~2.5–10 μg/mL in culture supernatant. All proteins appeared at their expected molecular mass in western blots under reducing and non-reducing conditions. Fc-diCBM40, however, was not stable (Fig. 1c). Attempts were made to further increase the yield of secreted proteins by expression in HEK293T *C1GalT1*-KO (knockout) cells that harbor truncated $O$-glycans[29], and also *SLC35A1*-KO cells that do not contain cell-surface sialic acid[30] (Supplementary Fig. 3a). Whereas wild-type HEK293T cells may express cell surface ligands that may bind the Fc fusion constructs, we hypothesized that the KOs would lack such ligands, potentially enhancing soluble protein concentrations. However, no such enhancement was observed (Supplementary Fig. 3a). Thus, all Fc-proteins were expressed using wild-type HEK293T and purified using 6xhis-tag prior to functional assays described below.

Enzyme activity and binding studies were performed to characterize PS1, H302A and Dead. In LC-MS/MS investigations, PS1 afforded the reaction product, Neu5Ac(α2-3)Gal(β1-3)GalNAc-$p$-nitrophenol (pNP), using acceptor substrate Gal(β1-3)GalNAc-pNP (TF-antigen), and donor CMP-Neu5Ac, similar to commercial human ST3Gal1 (Fig. 1d, and Supplementary Fig. 3b). Enzymatic activity was ablated in H302A and Dead. In cytometry studies, H302A displayed strong sialic acid-dependent binding to multiple epithelial cell types: kidney HEK293T, pancreatic COLO357-FG, and lung Calu-3, with particularly strong binding to the highly metastatic pancreatic cancer cells (Fig. 1e, and Supplementary Fig. 3c, d). Compared to H302A, Fc-CBM40 exhibited lower binding for HEK293T and COLO357-FG as might be expected for carbohydrate binding modules, and Dead was a non-binder. The sialic acid-binding property of H302A was similar to that of MALII, which binds α2,3-sialylated $O$-glycans[31], and it was distinct from PNA, which recognizes the TF-antigen. Compared to H302A, the sialic acid-dependent binding property of PS1 was weaker, with this dependence being more prominent at high protein concentrations (Fig. 1e, f). H302A and PS1 binding increased monotonically with lectin concentration up to 20 μg/mL (Fig. 1f). Overall, the introduction of the H302A mutation into pST3Gal1 led to a GBP that prefers to bind the product of the original enzymatic reaction.

### Specific binding of H302A to sialyl core-2 $O$-glycan

To define the binding specificity of H302A, a panel of isogenic HEK293T knockouts were created using CRISPR-Cas9 technology. This includes previously established *MGAT1*-KO and *C1GalT1*-KO cells lacking complex $N$-glycans and core-1 $O$-glycans respectively[29], *SLC35A1*-KO lacking the CMP-Neu5Ac transporter necessary for synthesis of sialoglycans, *GCNT1*-KO lacking core-2 $O$-glycans, and *ST3Gal1*-KO (Fig. 2a). Two different sgRNA (single-guide RNA) targeting each of these genes were cloned

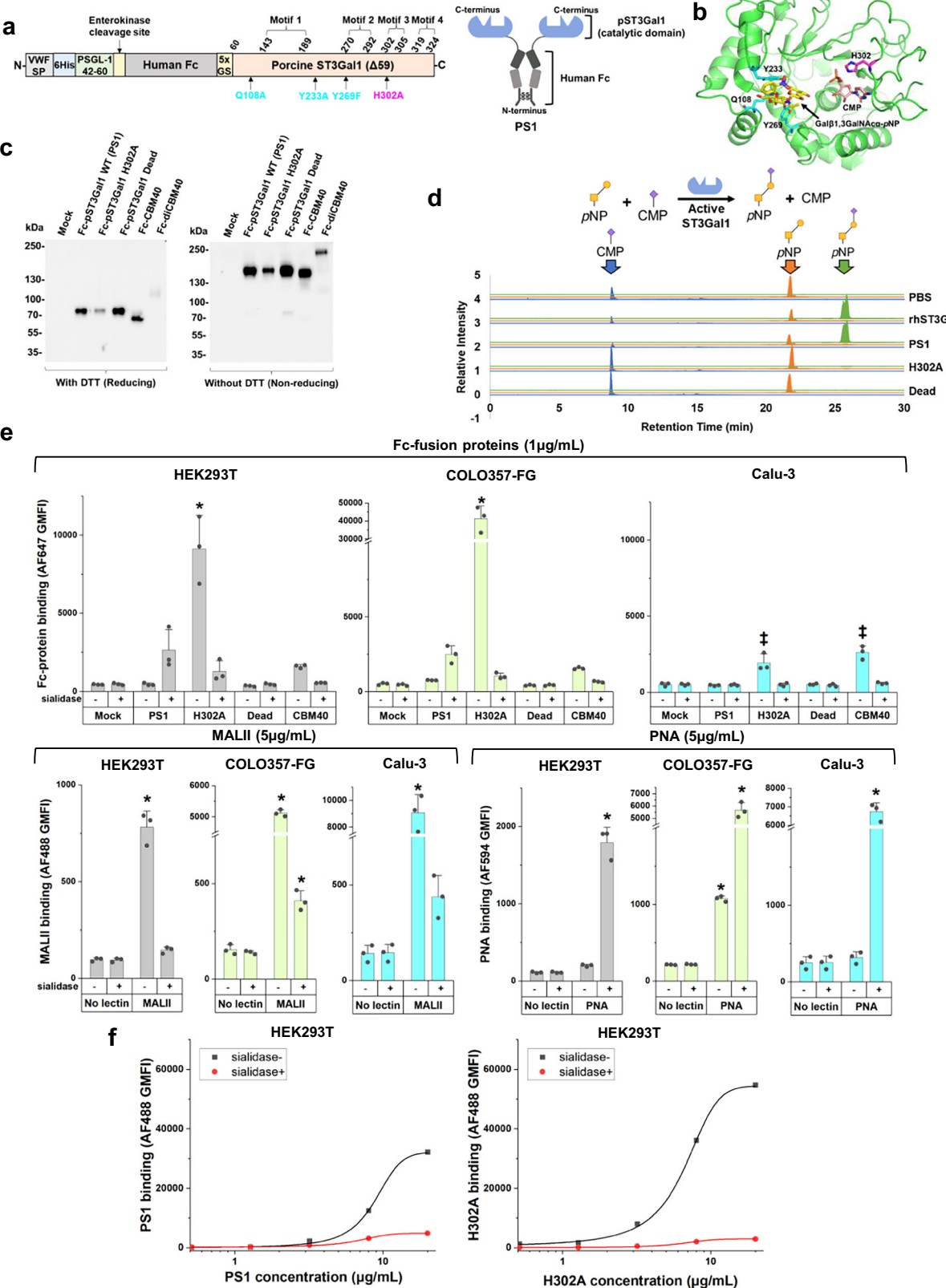

into the pX330 vector, mixed and transfected into HEK293T cells. Isogenic clones derived from this process commonly contained excisions between the two sgRNA target sites (Fig. 2b). As expected, *MGAT1*-KO reduced PHA-L and ECL binding consistent with the lack of formation of complex *N*-glycans and the reduced prevalence of the Gal(β1-4)GlcNAcβ epitope, respectively (Fig. 2c). *C1GalT1*-KO had low MAL-II and PNA

binding due to the absence of the TF-antigen. *ST3Gal1*-KO had reduced MAL-II binding on *O*-linked glycans. *SLC35A1*-KO had high ECL and PNA binding, and low MAL-II binding due to the absence of terminal sialic acid.

Among Fc-fusion proteins, H302A did not bind *C1GalT1*-KO, *SLC35A1*-KO, and *ST3Gal1*-KO but it bound *MGAT1*-KO similarly to wild-

**Fig. 1 | Fc-pST3Gal1 H302A mutant exhibits sialic acid dependent binding but lacks enzyme activity. a** Fc-pST3Gal1 (PS1) has four structural motifs. Amino acids shown in cyan (Q108A, Y233A, and Y269F) and magenta (H302A) were mutated in Dead and H302A mutants, respectively. **b** pST3Gal1 co-crystalized with CMP and Galβ1,3GalNAcα-*p*NP (PDB: 2WNB). Protein is in green, with selected residues and ligands shown as sticks. CMP is orange, Galβ1,3GalNAcα-*p*NP is yellow, amino acids mutated in Dead mutant are cyan, and H302A is magenta. **c** Western blots of Fc-fusion proteins expressed using HEK293T cells under reducing and non-reducing conditions. Detection was performed using anti-human Fc specific IgG. **d** LC-MS/MS analysis of ST3Gal1 enzymatic activity. Recombinant hST3Gal1 served as positive control. H302A and dead mutants did not yield product (N = 1). **e** Lectin binding to HEK293T (gray bars), COLO357-FG (green bars) or Calu-3 (blue bars) cells, with or without sialidase treatment, measured using flow cytometry. Cells were incubated with 5 μg/mL fluorescent MALII, PNA, or 1 μg/mL purified Fc-fusion proteins pre-complexed with AF647-conjugated anti-human Fc specific IgG. Data are mean ± STD (N = 3 biological replicates). *$p < 0.05$ with respect to all other samples; ‡$p < 0.05$ with respect to all other samples except samples marked by ‡ are not different from each other. *P* values calculated using one-way ANOVA followed by Tukey post-test. **f** Dose dependent binding of PS1 and H302A pre-complexed with AF488-conjugated anti-human Fc specific IgG. HEK293T cells treated with or without sialidase were used. Data are mean (N = 2 biological replicates). Source data including exact *P* values. Created in BioRender. Neelamegham, S. (https://BioRender.com/tigz92o).

type HEK293T cells (Fig. 2d). Additionally, this lectin displayed reduced binding to *GCNT1*-KO, which lacks core-2 *O*-glycan formation. The binding profile of H302A was markedly different from MALII in this core-2 dependence, suggesting that H302A is a novel GBP. PS1 exhibited similar binding specificity as H302A although the sialic acid dependence was only partial. CBM40 only showed partial sialic acid-dependence. Dead did not bind any of the cell types. Overall, H302A displayed unique binding to sialyl core-2 *O*-glycans, unlike other GBPs.

## Binding specificity of H302A confirmed using glycan array

We performed glycan microarray analysis with the CFG5.5 microarray as it contains a broad spectrum of 562 glycan structures (Fig. 3a). Here, both PS1 and H302A were observed to specifically bind only *O*-glycan epitopes and not epitopes related to *N*-glycans and glycolipids (Fig. 3b, c, and Supplementary Fig. 4a, b). Whereas PS1 predominantly bound non-sialylated core-2 *O*-glycans, it displayed less binding to non-sialyl core-1 *O*-glycans. H302A, in contrast, only recognized two glycans, sialyl core-2 *O*-glycan (**561**) and sialyl 6-sulfated *O*-glycan (**237**). The binding of both PS1 and H302A was quite specific in that they did not recognize several closely related 3-*O*-sulfated (**29**), fucosyl (**61, 62**), monosialyl (**134, 135, 219, 220**), disialyl (**238, 239**) TF antigens, sialylated Tn antigen (**240**), and core-2 *O*-glycans with sialic acid cap on core-2 branch (**281, 309, 326, 560**) (Fig. 3d, Supplementary Data 1). Furthermore, these lectins also did not bind additional sialoglycans (**233, 245**) that had structures very similar to (**237**). The binding specificity of H302A and PS1 was also distinct from PNA and MALII, which displayed much broader binding specificity, thus highlighting the important advantage of engineering GTs for GBP applications (Supplementary Fig. 4c–f).

Given the strong binding of H302A to 6-*O*-sulfated GalNAc (**237**), we investigated if the lectin may bind physiological sulfated glycans. Such structures are synthesized by carbohydrate sulfotransferases belonging to the CHST and Gal3ST families[32–34], but none of these enzymes are currently known to have 6-*O*-sulfotransferase activity towards GalNAc. To examine this in a cellular context, we treated promyelocytic HL60 cells with sodium chlorate, a potent ATP-sulfurylase inhibitor that blocks sulfated epitope biosynthesis (Fig. 3e). This treatment did not impact PS1 and H302A binding to leukocytes, but it reduced P-selectin Fc-fusion protein binding, which is consistent with the known role of protein tyrosine sulfation in leukocyte adhesion[35]. The data suggest that H302A likely only binds sialyl core-2 *O*-glycans in a cellular context. 6-*O*-sulfated sialyl *O*-glycans such as (**237**) likely represent non-physiological glycans present on the CFG5.5 microarray.

## CRISPR screen reveals genes regulating PS1 and H302A binding

Only a limited repertoire of natural glycans is presented on the CFG microarray, and their presentation on solid-support is non-physiological. We thus performed forward genetic screens using a previously developed HL60 glycoCRISPR cell library, to determine the glycogenes regulating lectin binding to cells. This knockout library allows the study of 347 glycogenes involved in human carbohydrate biosynthesis[30,36]. To achieve this, anti-human IgG Fc specific antibodies were conjugated onto epoxy-functionalized magnetic beads (Fig. 4a). HL60 glycoCRISPR library cells pre-incubated with PS1 or H302A were then mixed with these beads (Fig. 4a). In this system, cells engaging Fc-fusion proteins are captured by the beads. The remaining cells that do not bind the GBPs remain free and were negative-selected. These negative-selected cells lack critical glycogenes necessary for H302A/PS1 binding. Three rounds of negative selection were thus performed to enrich the non-binding population. In the case of H302A, a distinct negative population was observed after the third sort (red bar in Fig. 4b). The signal decrease for negative-selected PS1 was less obvious compared to the starting cells, likely due to lower binding affinity (Fig. 4c). In both cases, similar sgRNA/glycogenes were responsible for loss of binding function, as determined using next-generation sequencing (NGS). This includes all enzymes involved in sialic acid biosynthesis (*NANS*, *GNE*, *CMAS*, and *SLC35A1*), core-1 *O*-glycan biosynthesis (*C1GalT1C1* and *C1GalT1*), and core-2 *O*-glycan biosynthesis (*GCNT1*) (Fig. 4b, c, right panel). Sialyltransferases *ST3Gal1* and *ST3Gal4* were also observed in the case of PS1. While these enzymes were detected in early rounds of the selection process in the case of H302A (see Source Data), their absence in the third round suggests incomplete penetrance of these enzyme, possibly due to overlapping specificity for these enzymes for the Type-III lactosamine chain (Galβ1-3GalNAcα)[37]. Overall, PS1 and H302A showed similar dependence for *O*-glycosylation specific genes, with some difference related to *ST3Gal* contributions. No conclusion could, however, be reached regarding the impact of lactosamine chain extension on the core-2 arm since multiple Gal and GlcNAc-transferases contribute to such synthesis.

## Mammalian surface display for high-throughput GBP engineering

Phage display[38] and yeast surface display[39] are commonly used for protein engineering, and similar high-throughput screening of mammalian GBPs requires the development of analogous mammalian surface display platforms. To anchor proteins on mammalian cell surface, we fused the cytoplasmic and transmembrane domains of the type-2 single-pass transmembrane protein, dipeptidyl peptidase 4 (DPP4), to the N-terminus of our Fc-protein constructs (Fig. 5a, b, sequence in Supplementary Fig. 2d).

Using this strategy, TM-PS1, TM-H302A and TM-Dead were robustly expressed on HEK293T surface (Fig. 5c, d). These surface displayed Fc-proteins, however, were unable to bind a fluorescent sialyl core-2-PAA-FITC polymer that was synthesized upon sialylation of commercially available core-2-PAA-FITC polymer using rhST3Gal1 and CMP-Neu5Ac. We reasoned that this may be due to *cis*-interactions between surface displayed GBPs and endogenous sialoglycans presented on the cell surface (Fig. 5c, [27,40,41]). Indeed, flow cytometry-based dual-color cell-cell interaction assays show that sialidase treatment of HEK293T cells expressing TM-PS1 and TM-H302A increases heterotypic binding of surface display cells to wild-type HEK293T ligand cells that bear their counter-receptor (Supplementary Fig. 5). This binding is blocked by sialidase treatment of the ligand bearing cells, and also upon surface expression of TM-Dead (instead of TM-PS1 and TM-

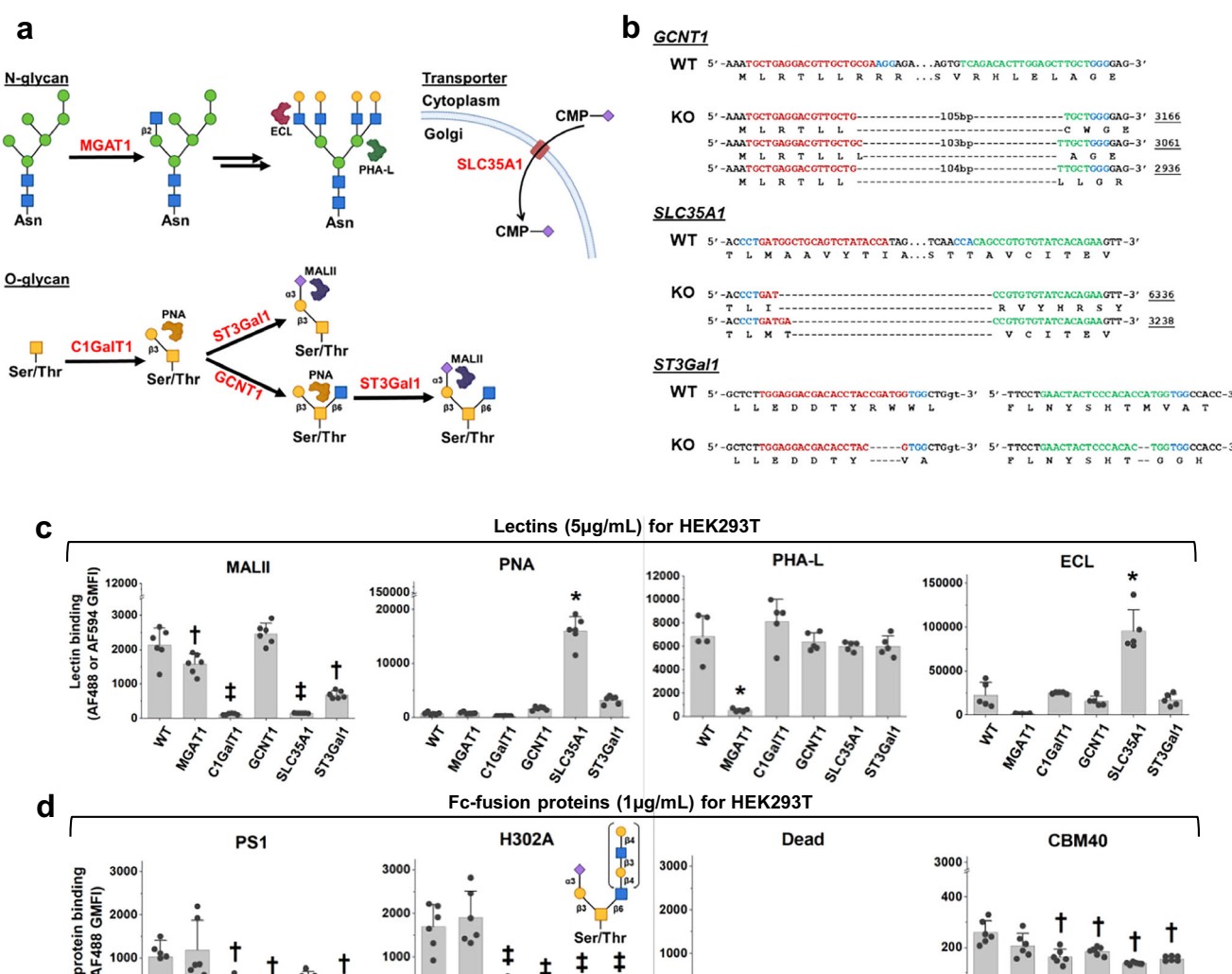

**Fig. 2 | H302A binds sialylated core-2 *O*-linked glycans. a** Isogenic HEK293T KO cells lacking selected glycan-processing enzymes or transporters were produced. **b** Next-generation sequencing (NGS) results for isogenic clones lacking *GCNT1*, *SLC35A1*, and *ST3Gal1*. Wildtype DNA and amino acid sequence corresponding to sense strand are shown above, with gene edited sequence below. Exons and introns appear in uppercase and lowercase, respectively. As two sgRNA were added simultaneously to create KOs, red and green represent the first and second editing regions, and blue presents the protospacer adjacent motif (PAM) on sense (*GCNT1* and *ST3Gal1* KO) or anti-sense strands (*SLC35A1* KO). The entire sequence between the sgRNA cut sites was excised in the *GCNT1* and *SLC35A1* KO. This resulted in a few different closely-resembling read sequences. The frequency of occurrence of each of these edit patterns is shown on the right side of the DNA sequence (underlined). **c** Flow cytometry binding assay using 5 μg/mL fluorescent lectins (MALII and PNA for *O*-glycans, PHA-L and ECL for *N*-glycans). HEK293T WT cells and panel of isogenic HEK293T KO cells were used. **d** Flow cytometry binding assay using 1 μg/mL purified Fc-fusion proteins pre-complexed with AF488-conjugated anti-human Fc specific IgG. Data are mean ± STD (N = 6 for (**c**, **d**), except N = 5 for PHA-L and ECL, biological replicates). *P* values calculated using one-way ANOVA followed by the Tukey post-test. Data show that H302A binds sialylated core-2 *O*-linked glycans. *$p < 0.05$ with respect to all other samples; $^{‡}p < 0.05$ with respect to all other samples except that samples marked by $^{‡}$ are not different from each other; $^{†}p < 0.05$ with respect to WT cells. Source data are provided in Source Data file. Created in BioRender. Neelamegham, S. (https://BioRender.com/tigz92o).

H302A). Consistent with this notion, sialyl core-2-PAA-FITC displayed robust binding to cells bearing TM-PS1 and TM-H302A upon sialidase treatment (Fig. 5d). This sialylated polymer, however, could not bind TM-Dead cells, as this is a non-binder.

To screen for TM-H302A variants with improved binding to sialyl core-2-PAA-FITC, two plasmid libraries were created, each targeting selected residues proximal to the sialic acid binding pocket of pST3Gal1 based on an AlphaFold molecular modeling (Fig. 5e, and Supplementary Fig. 6a). Lib1 constitutes a library with 440 mutants focused on amino acids prior to motif 2 in the protein primary sequence, while Lib2 contains 1240 mutants in later residues (Fig. 5a). While a vast majority of constructs contained the H302A mutation, the

library also includes members where H302 is modified to other amino acids besides alanine. Amino acid representation in all libraries was verified using NGS (Supplementary Fig. 6b). Lentivirus corresponding to both libraries was produced, and these were used to transduce HEK293T *SLC35A1*-KO cells that lack endogenous sialoglycans. Virus was applied at low multiplicity of infection (MOI < 0.25) in order to achieve ~1 mutant/cell (Supplementary Fig. 7a). In cytometry binding assays, the Lib2 cells exhibited greater binding to sialyl core-2-PAA-FITC, compared to either TM-H302A or Lib1 cells (Fig. 5f). Two rounds of FACS sorting were thus performed with Lib2 to select for cells with superior sialyl core-2-PAA-FITC binding properties (Supplementary Fig. 7b). One round of sorting was performed for Lib1. NGS analysis

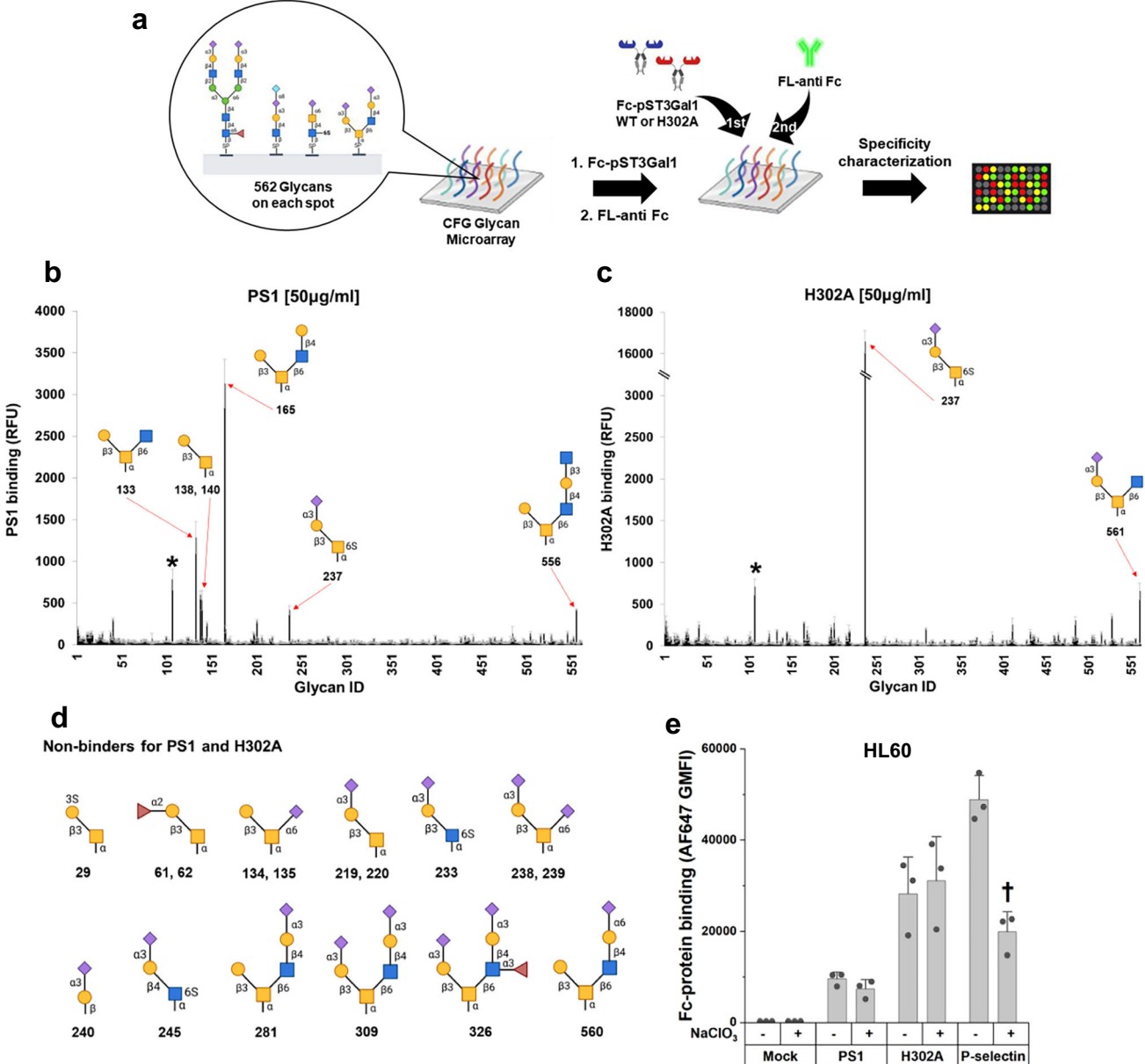

**Fig. 3 | H302A displays strict binding specificity for sialylated core-2 motif.**
**a** Schematic of glycan microarray study performed using CFG 5.5 with 562 immobilized glycans. PS1 or H302A were added to microarray at either 5 µg/mL or 50 µg/mL, followed by detection using fluorescent anti-human Fc specific IgG (FL-anti Fc). **b, c** Lectin binding quantified using relative fluorescence units (RFU) for PS1 (panel b) and H302A (panel c). Data are mean ± STD (N = 6 using six different microarrays). Red arrows with glycan symbol and number represent strong binders. * Glycan 107 is non-specific binder, as its binding is Fc-protein dose independent. **d** Non-binders of H302A and PS1 are shown. Neither Fc-fusion proteins recognized α2,3-sialyl core-1 *O*-glycans (219 and 220), α2,6-sialyl core-1 *O*-glycans (134, 135, 238, and 239), or core-2 *O*-glycan with α2,3-sialyl LacNAc chains (281, 309, 326, and 560), or a host of

*N*-glycan and glycolipid structures. **e** HL60 WT cells were cultured with sodium chlorate to prevent sulfation, prior to measuring the binding of 1 µg/mL Fc-fusion protein, pre-complexed with AF647-conjugated anti-human Fc specific IgG. P-selectin Fc binding, but not H302A or PS1 binding, was chlorate sensitive. Results confirm sialylated core-2 *O*-glycan binding specificity of H302A. Glycan structures are depicted using the Symbol Nomenclature For Glycans standard[68]. Data are mean ± STD (N = 3, biological replicates). *P* values calculated using two-tailed unpaired Student's *t*-test. †*p* < 0.05 with respect to no sodium chlorate treatment. Source data including exact *P* values are provided in Source Data file. Created in BioRender. Neelamegham, S. (https://BioRender.com/tigz92o).

---

revealed several mutations in Lib2 with high positive enrichment scores (Fig. 5g, and Supplementary Data 2), while Lib1 selection did not reveal any candidate hits (Supplementary Fig. 7c). The Lib2 mutations with enhanced sialyl core-2 binding were centered at residues S271 - T272 and A312 - F313 of the full protein.

## Characterization of sCore2 glycan binding protein
The top six mutation candidates from the above screen were individually cloned and expressed as soluble Fc-proteins. In western blots,

all H302A/A312X/F313X mutants expressed well, but the H302A/S271X/T272X clones did not express (Fig. 6a). Studies were performed with the panel of isogenic HEK293T-KOs to characterize lectin binding properties. These data showed that all H302A/A312X/F313X mutants retained the sialylated core-2 *O*-glycan binding properties of H302A (Fig. 6b). Among the mutants, H302A/A312I/F313S displayed stronger sialic acid-dependent binding compared to the original H302A (Fig. 6c, Supplementary Fig. 7d). This was further confirmed by performing dose dependence binding assays (Fig. 6d). Due to distinct binding to

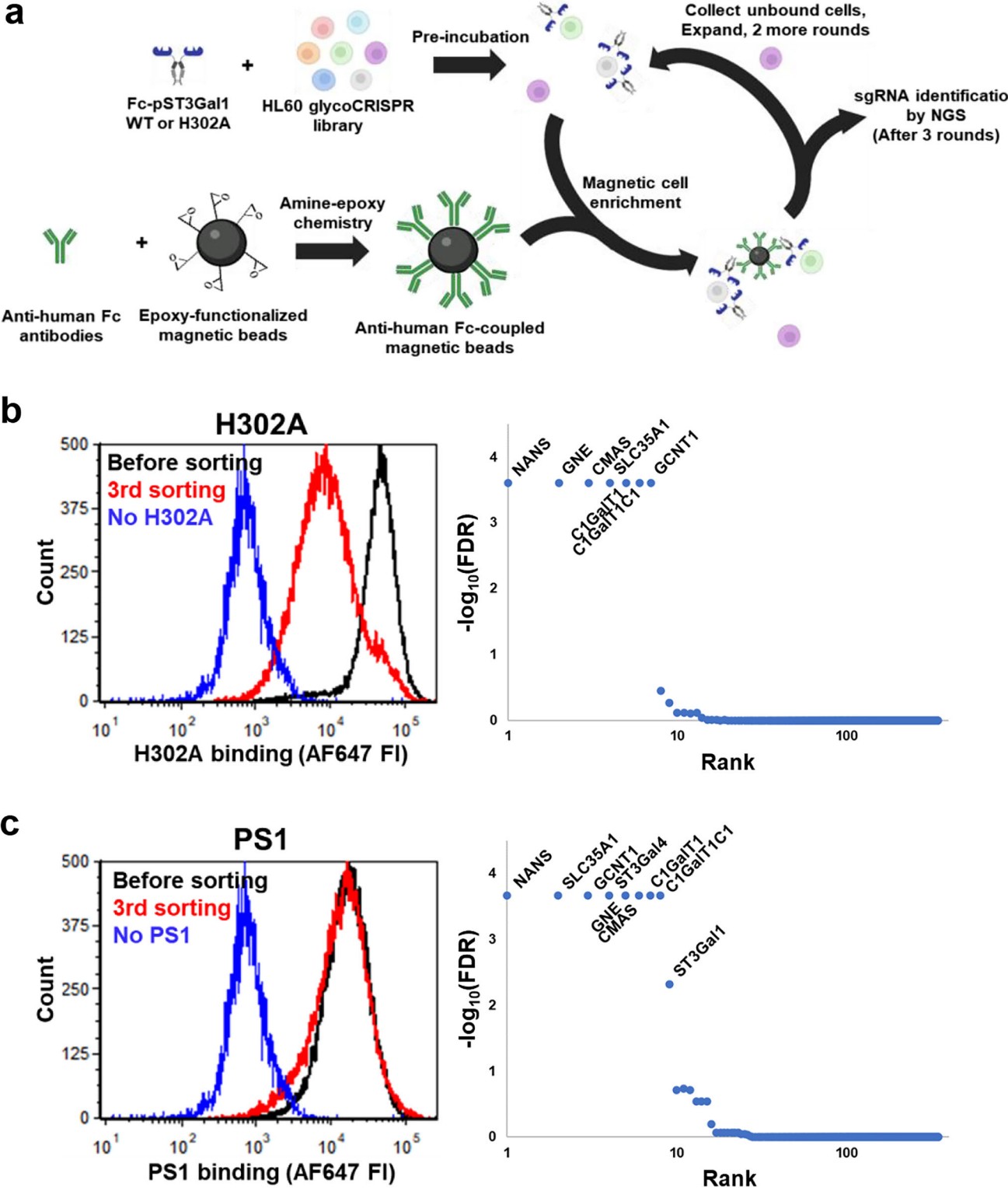

**Fig. 4 | Forward genetic CRISPR-screen identifies glycoEnzymes regulating PS1 and H302A binding. a** Anti-human Fc antibodies were covalently coupled to epoxy-functionalized magnetic beads. The beads were added to HL60 glycoCRISPR library cells that were pre-incubated with PS1 or H302A. Cells binding the Fc-proteins were captured onto magnetic beads, whereas the non-binders remained in solution. These unbound cells were isolated, expanded, and subjected to two more rounds of magnetic enrichment. Genomic DNA was isolated and sequenced to determine sgRNA and corresponding genes regulating Fc-protein binding. **b**, **c** Flow cytometry histograms on left side show the HL60 glycoCRISPR library binding profiles for H302A (**b**) and PS1 (**c**) before sorting, after 3rd sort and negative control without Fc-protein. SgRNA enrichment plots on right side identify genes from the 3rd sort, whose depletion by CRISPR-Cas9 also reduced cell-lectin conjugate capture by magnetic beads for H302A (**b**) and PS1 (**c**). Glycogenes critical for lectin binding (FDR < 0.005) were labeled in the figure. Source data are provided in Source Data file. Created in BioRender. Neelamegham, S. (https://BioRender.com/tigz92o).

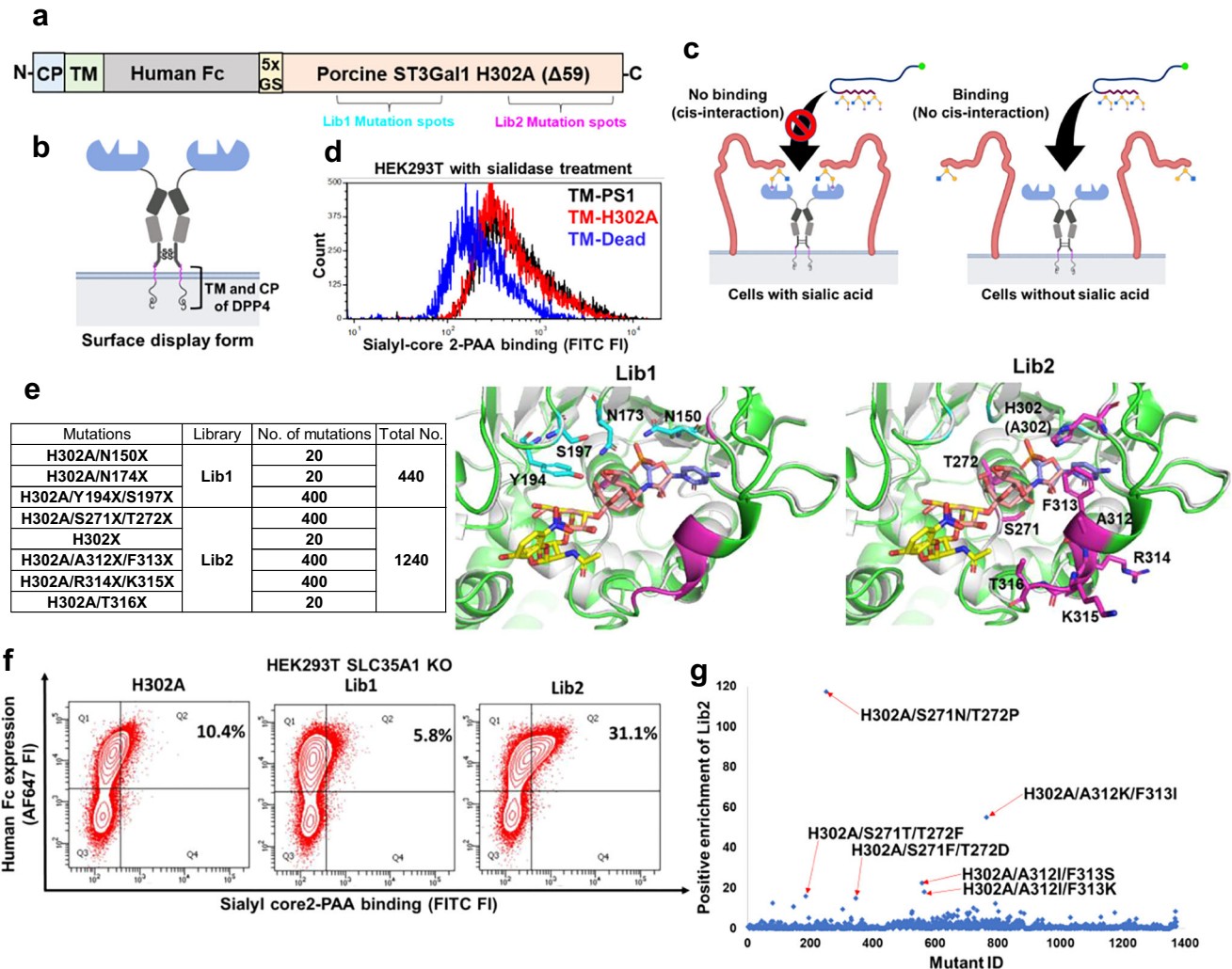

**Fig. 5 | Human surface display platform enriches H302A binding properties.**
**a, b** Fusing the cytoplasmic and transmembrane domain of type-II transmembrane protein DPP4 at the N-terminus of Fc-proteins (H302A in this example) enabled a mammalian surface display platform. Amino acids mutated to create Lib1 and Lib2 were shown in cyan and magenta, respectively in (**a**). **c, d** HEK293T WT cells were transfected to surface display TM-PS1, TM-H302A and TM-Dead. Cells were treated with sialidase to eliminate *cis*-interactions (**c**). AF647-conjugated goat anti-human Fc-specific IgG was used to monitor Fc expression level. Cells with high Fc expression were gated and used in the histogram in panel d as a measure of lectin binding with sialyl core-2 PAA-FITC. **e** Amino acid residues mutated in Lib1 and Lib2 are shown in Alphafold model of pST3Gal1 (green) and its crystal structure (PDB: 2WNB, gray) co-crystalized with CMP (purple stick) and Galβ1,3GalNAcα-*p*NP (yellow stick). These residues, which are represented as cyan (Lib1) and magenta (Lib2) sticks, were mutated for optimizing the glycan binding properties of H302A. Neu5Ac superimposed from 5FRE was shown as orange sticks. **f** Sialyl core-2 PAA-FITC binding to HEK293T *SLC35A1*-KO cells transduced to express TM-H302A, Lib1, and Lib2. **g** Identification of top-6 mutants in Lib2 that are superior binders of sialyl core-2 PAA-FITC. Source data are provided in Source Data file. Created in BioRender. Neelamegham, S. (https://BioRender.com/tigz92o).Source Data

the sialyl core-2 epitope, H302A/A312I/F313S pST3Gal1 Fc-protein was named sCore2.

To further characterize sCore2 binding, negative-selection studies were performed using the HL60 glycoCRISPR library using the method in Fig. 4, only using flow cytometry sorting for negative selection. After two rounds, sCore2 binding to the sorted cells was low compared to wildtype (Fig. 6e). sgRNA enriched following the second sort targeted genes related to sialic acid (*CMAS, GNE, NANS, SLC35A1*), core-1 (*C1GalT1C1, C1GalT1*), and core-2 (*GCNT1*) *O*-glycan biosynthesis. *N*-glycosylation or glycolipid biosynthesis genes were absent. Similar to H302A, whereas sgRNA against both *ST3Gal1* and *ST3Gal4* were enriched following one round of flow sorting, only *ST3Gal4* remained after the second round. To elaborate on this, we tested the ability of sCore2 and additional lectins to bind previously developed HL60 *ST3Gal4*-KO cells[42] (Fig. 6f). Here, partial reduction in both H302A and sCore2 binding was observed, but such binding was abolished for PS1. Based on these observations and previous work showing that human ST3Gal4

can sialylate Gal(β1-3)GalNAcα-*O*-Benzyl substrate[37,43], it seems possible that both ST3Gal1 and ST3Gal4 may contributes to the formation of the sialyl core 2 *O*-glycan structure in HL-60. Overall, the binding specificity of sCore2 mimicked H302A, only showing greater sialic acid dependence.

## Expression of sialyl core-2 *O*-glycans on selected blood cells and tissue

sCore2 was applied in spectral flow cytometry studies to investigate the *O*-glycan profiles of human blood cell types. To this end, the Cytek immunoprofiling kit was modified to include channels for fluorescent lectins and to expand the coverage of myeloid sub-populations that were otherwise missed. Three related *O*-glycan specific lectins, sCore2, MALII, and PNA, were applied. Control studies confirmed minimal competition among these reagents for different blood cell populations, likely because only a small fraction of glycan epitopes on cells are bound by these lectins (Supplementary Fig. 8a). Using this

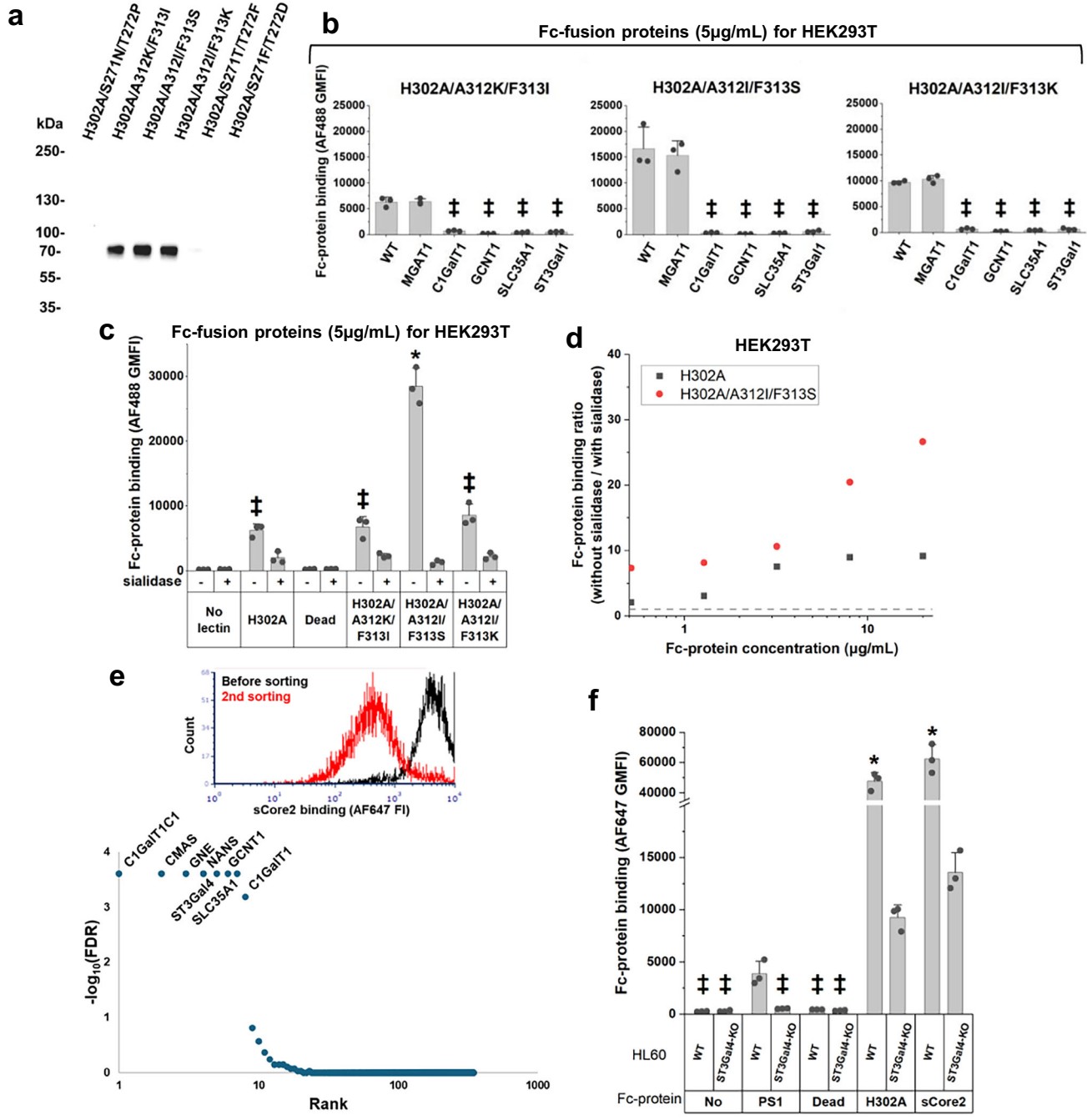

**Fig. 6 | sCore 2 is superior to H302A in binding sialyl core-2 glycans. a** Western blots of expressed mutants identified by library screening. Three of them expressed well. **b** 5 µg/mL purified Fc-fusion proteins pre-complexed with AF488-conjugated anti-human Fc specific IgG were incubated with a panel of HEK293T cells (wild-type and KO). All Fc-proteins bound in a sialyl core-2 dependent manner. (**c**) 5 µg/mL purified Fc-fusion proteins were pre-complexed with AF488-conjugated anti-human Fc specific IgG, and incubated with HEK293T cells, with or without sialidase treatment. H302A/A312I/F313S (sCore2) bound HEK293T cells at levels 3-fold greater than H302A. **d** Dose dependent binding of H302A and H302A/A312I/F313S (sCore2) pre-complexed with AF488-conjugated anti-human Fc specific IgG to HEK293T cells. sCore2 showed stronger sialic acid binding preference. Dotted line represents the ratio of 1. **e** Glycogene CRISPR screen showing the glycogenes regulating sCore2 binding. **f** Binding of glycan binding proteins to HL60 (wild-type and *ST3Gal4*-KO). All data are mean ± STD (N = 3 for all, except N = 2 for dose dependence study, all biological replicates). *P* values were calculated using one-way ANOVA followed by Tukey post-test. *$p < 0.05$ with respect to all other samples; ‡$p < 0.05$ with respect to all other samples except that samples marked by ‡ are not significantly different from each other. Source data including *P* values are provided in Source Data file.Source Data

approach, we discerned the lectin binding profiles of 24 different blood cell subtypes as depicted using tSNE (t-distributed stochastic neighbor embedding) plots (Fig. 7a, b, and Supplementary Fig. 8b, c). Here, PNA binding was largely restricted to classical and non-classical monocytes, neutrophils, and eosinophils, but not basophils and lymphoid cells. MAL-II binding was almost inverse, suggesting that the extent of terminal sialylation of the Neu5Acα2-3Galβ1-3GalNAcα arm may be less on myeloid cell populations compared to lymphoid cells. Only naïve B-cells were low for MAL-II and PNA, and these findings were consistent across donors (Supplementary Fig. 8c).

The binding profile of sCore2 was different from other lectins in that it bound myeloid cells and terminal effector T-cells rather than

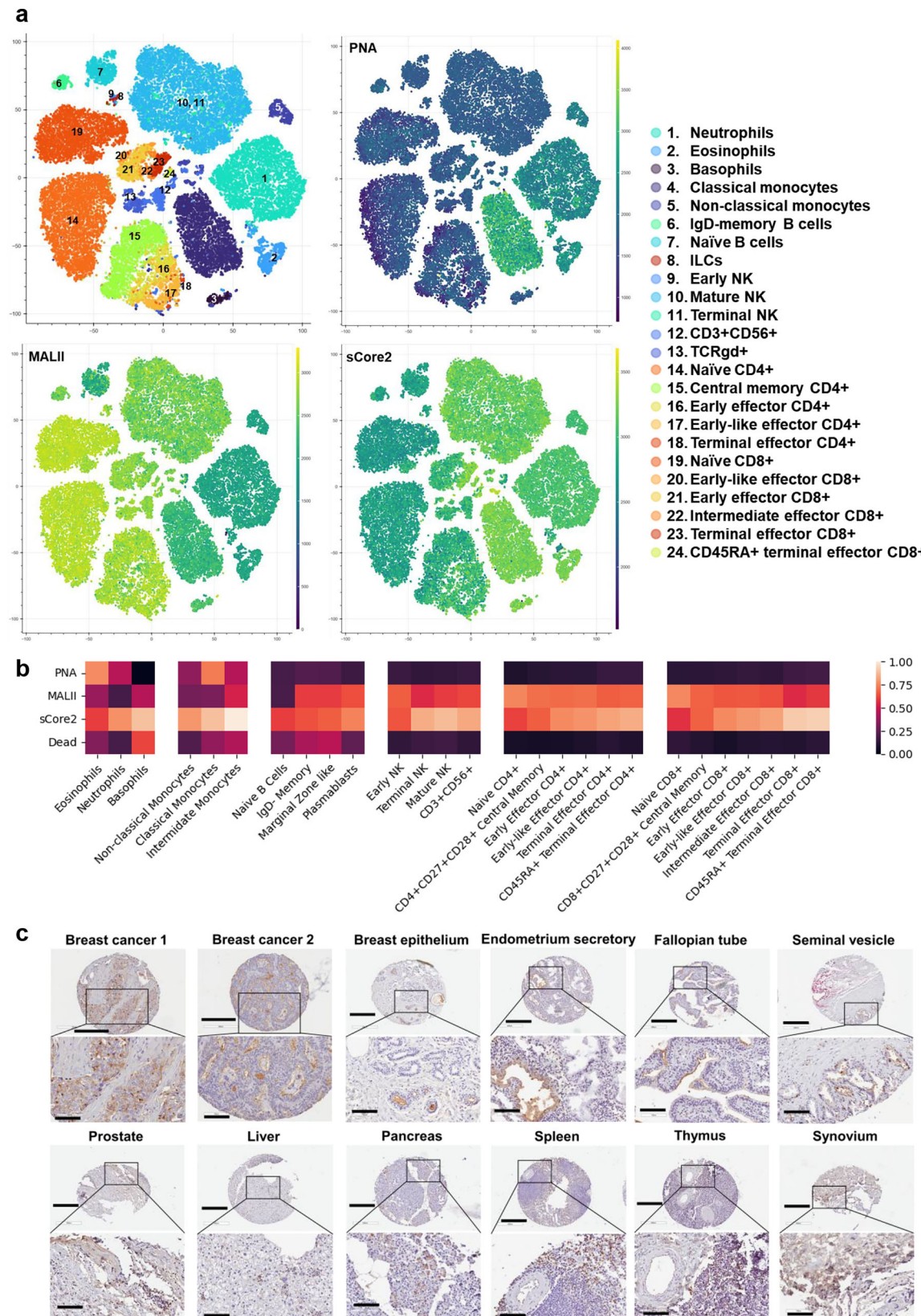

1. Neutrophils
2. Eosinophils
3. Basophils
4. Classical monocytes
5. Non-classical monocytes
6. IgD-memory B cells
7. Naïve B cells
8. ILCs
9. Early NK
10. Mature NK
11. Terminal NK
12. CD3+CD56+
13. TCRgd+
14. Naïve CD4+
15. Central memory CD4+
16. Early effector CD4+
17. Early-like effector CD4+
18. Terminal effector CD4+
19. Naïve CD8+
20. Early-like effector CD8+
21. Early effector CD8+
22. Intermediate effector CD8+
23. Terminal effector CD8+
24. CD45RA+ terminal effector CD8+

immature T- or B-cells. Besides terminal CD4+, and CD8+ T-cells, increased binding was also noted for mature and terminal NK cells compared to early NK populations. As MAL-II binding decreased with such cell differentiation, the data suggest a transition from mono- and di-sialyl *O*-glycans to sialyl core-2 structures upon lymphocyte differentiation. Low binding of sCore2 to naïve and central memory T-cells,

and high binding to effector T-cells is consistent with observations of increased GCNT1 expression upon T cell maturation[44,45]. To the best of our knowledge, this is the first report that NK-cell differentiation is also correlated with the sialyl core-2 *O*-glycan levels. With regard to B-cells, it is proposed that *GCNT1* mRNA levels are reduced upon B-cell maturation from naïve to memory subtypes[46]. However, our data with

**Fig. 7 | Application to cell-surface glycan detection of human peripheral blood samples and tissue sections from variety of organs. a** tSNE plots showing the binding of PNA, MALII and sCore2 to different human blood cell types, immuno-profiled using 23 antibodies. Representative data are shown for a single donor. Legends show numbers and color labels for each cell population in tSNE plot. **b** Heatmap showing the average normalized binding of each lectin to different human blood cell types, based on studies with two donors. Heatmaps are arranged based on cell lineage. Some cell types from heatmap do not appear in tSNE plots as these are rare populations, with small number of cells. **c** Human normal tissue microarray and breast cancer tissue analysis using sCore2. Staining of selected cores are shown, along with higher magnification tissue sections. Data for negative controls using Dead Fc-protein and upon sialidase treatment are presented in Supplementary Information. Results show cell specific staining properties of sCore2. Images are representative of two independent experiments. Main figure scale bar = 400 µm (except 300 µm for breast tumors). Magnified image scale bar = 100 µm. Source data are provided in Source Data file.Source Data

sCore2 suggest that these transcript level changes may not quantitatively contribute to core2 branching patterns on B-cells. As expected, Dead showed low binding to all peripheral blood cell populations, compared to the binding of sCore2 (Supplementary Fig. 8d).

We used sCore2 to detect the sialyl core-2 O-glycan epitope on 40 different normal human tissue on a tissue microarray and a limited number of cancer tissue (Fig. 7c, and Supplementary Fig. 9a, b, Supplementary Data 3). In all cases, negative control studies performed using the Dead mutant and upon sialidase treatment showed minimal signal. In comparison to normal breast tissue, staining was moderate to strong (greater than 50%) in invasive breast ductal carcinoma within the glandular or follicular regions, particularly in the apical lumen and associated microlumens. Additionally, the sialyl core-2 O-glycan expression was observed in the glandular cytoplasm of the normal endometrium, apical cytoplasm of fallopian tubes, and cytoplasm of pancreas, and inflammatory cells in liver. In the case of spleen, staining was weak in the cytoplasmic lymphoid white pulp, but stronger in the red pulp sinusoidal cells (>80%). sCore2 staining was also weakly observed in thymus and synovium, in ~50% of capillary endothelium of the cardiac muscles, and 70% of the granular neuropils of the brain. Overall, we noted sCore2 staining on tumor cells, blood cells and specific cell types from selected normal organs. However, staining was not observed in cells from the oral and salivary regions (salivary glands, tonsils), lungs, soft tissue (cartilage, skeletal muscles, intestines, uterus), gastrointestinal (gastric and colon mucosa) and urological tract (kidney, bladder).

To determine if sialyl core-2 epitope expression correlates with Core-2 N-acetylglucosaminyltransferase, selected tissues were stained using sCore2 along with anti-GCNT1 Ab. Partial overlap in staining was observed in breast cancer and normal spleen tissues (Supplementary Fig. 10), although anti-GCNT1 showed broader staining. The distinct patterns suggest that while GCNT1 may be necessary for sCore2 binding, it is not alone sufficient as the binding epitope of sCore2 is more complex.

## Discussion

The paper introduces protein engineering strategies to convert mammalian GTs into GBPs that recognize the substrate, product or related carbohydrates. This provides a strategy to develop lectins-by-design, rather than relying solely on screening strategies. Importantly, the work resulted in lectins that bind sialylated core-2 glycans on many cell and tissue types. This overcomes a challenge with raising antibodies that cannot be easily generated against core-2 O-glycans as they are naturally occurring in mammalian cells. Indeed, while a mouse anti-CD43 mAb, 1B11, has been proposed to bind core-2 O-glycans[47], its protein dependence restricts its use to blood cells which express CD43.

In studies that contrast PS1 with H302A, we observed that both Fc-fusion proteins function as lectins that bind core-2 O-glycans. These lectins prefer to bind core-2 O-glycans rather than linear core-1 chains, consistent with the known substrate preference for ST3Gal1 which transforms core-2 acceptors ($K_M = 8$ µM) more readily compared to core-1 substrates ($K_M = 50$ µM)[37]. In cell-based assays, PS1 preferentially bound to sialoglycans at high concentrations, but it also bound non-sialylated carbohydrates at lower concentrations and in glycan

microarray studies. These observations suggest the importance of His302 in regulating lectin binding. Since this residue naturally interacts with CMP-Neu5Ac to catalyze chemical reactions, mutating it to alanine abolishes this transformation. This modification may then alter the side chain orientations of other amino acids in the binding pocket, like Y194, promoting hydrogen bonds with the carboxyl residue of sialic acid (Supplementary Fig. 11a). As His302 is highly conserved in many mammalian GT29 family sialyltransferases, it would be interesting to implement the same or similar mutations on other sialyltransferases also, as this may afford additional specific, sialic acid-binding lectins.

The broad binding interface of PS1 and H302A is different from that of other sialoglycan-binding proteins like MAL-II, SNA (*Sambucus nigra agglutinin*), influenza hemagglutinins[48,49] and Siglecs[50], none of which display core-2 dependent ligand recognition. This is consistent with the observation that H302A requires two different carbohydrate motifs for high affinity binding: Neu5Ac(α2-3)Gal and Core-2. The crystal structure of pST3Gal1 affords space to accept core-2 GlcNAc, including forming potential hydrogen bond interactions (Supplementary Fig. 11b). However, the bulky nature of Neu5Ac may prevent inclusion of Neu5Acα2,6GalNAc due to amino acid clashes (Supplementary Fig. 11c). Additionally, two arginine residues, R109 and R268, around the C6 hydroxy group of GalNAc may form electrostatic interactions if a sulfate substituent replaces GlcNAc, and this may explain H302A binding to 6-O-sulfated sialyl TF-antigen in the glycan microarray studies (Supplementary Fig. 11d). However, such structures are not yet reported in mammals, and human sulfotransferases were not identified to be critical in our CRISPR genetic screen. Indeed, several other sialoglycan-binding proteins such as MALII and Siglec families also recognize sulfated glycans[5,50,51], which may also be driven by similar electrostatic interactions. Thus, mutations in R109 and R268 may allow further tuning of H302 specificity. While this study established the binding preference of H302A for sialyl core-2 O-glycans, the repertoire of glycans on the glycan microarray was insufficient to explain the types of LacNAc chain extensions on the core-2 arm that are tolerated by this lectin. Overall, the core-2 O-glycan binding preference of PS1 and H302A and enhanced binding specificity of H302A for α(2-3) sialylated glycans are consistent with available structural evidence and enzyme kinetics data.

We established a mammalian cell surface display platform for high-throughput screening of GT variants. In this context, indeed, mammalian cell surface display platforms have been used previously to present chimeric antigen receptors for immunotherapy[52], to select for antibodies following surface display[53,54], for screening viral proteins[55] and engineering leguminous lectins[56]. While there is merit in each of these applications, most mammalian surface-display methods focus on the display of type-I transmembrane proteins, while the engineering of glycosyltransferases requires type-II transmembrane display. Searching through literature, we noted one example of type-II transmembrane display that was used to screen for short cysteine-rich peptides[57]. However, to our best knowledge, this manuscript is the first example where Type-II transmembrane surface display has been used for engineering glycosyltransferases.

Using the Type-II surface display platform, we focused on residues proximal to the sialic acid binding pocket including the

disordered loop[21], which upon folding interacts with the bound ligand. These studies identified H302A/A312I/F313S (sCore2) as a superior sialyl core-2 binder compared to H302A alone. As A312 and F313 are missing in the original crystal structure[21], AlphaFold modeling was performed and this suggests that the large phenyl group of F313 may clash with the bound ligand. The substitution F313S may reduce bulkiness, promoting hydrogen bond formation with Neu5Ac residues, such as C5-acetamide (Supplementary Fig. 11e, f). The development of sCore2 validates the overall experimental focus on GTs, including the rationale for the directed evolution campaign. Looking forward, it is possible that any mammalian GTs could be expressed using this system, and this could be a useful platform for both engineering new GBPs and high-throughput screening of GTs. In addition, starting from a single GBP (like H302A), it may be possible to diversify the library to create additional members that specifically bind related *O*-glycan epitopes. The availability of well-defined chemical entities for screening and additional glycogene CRISPR KO-libraries can accelerate such development. Whereas standard site-directed mutagenesis is used in our screening approach, more modern mammalian cell-based directed evolution, such as TRACE[58], CRISPR-X[59], error-prone PCR[13], and adenovirus-based continuous evolution[60] may also be used to further enhance this approach. This would result in a broader set of mutants.

New and specific lectins are needed to recognize clinically significant carbohydrate epitopes, as glycan transformation commonly accompanies metabolic disorders[2,4], as gene expression alone does not directly correlate with carbohydrate alterations[1] and as glycoscience mass spectrometry techniques lag behind clinical practice[61]. In this context, sCore2 binds sialyl core-2 *O*-glycans on peripheral blood cells efficiently, particularly selected myeloid cell and effector lymphoid cell populations. This reagent also binds distinct cell types in normal human tissue microarrays, with data suggesting predilection to bind cancer cells. This was observed for breast cancer tissue compared to normal breast tissue and highly metastatic COLO357-FG pancreatic cells. Indeed, levels of some β1-6GlcNActransferases like *GCNT1* are significantly upregulated in a number of TCGA (The cancer genome atlas) cancers like kidney chromophobe (10.7-fold, adjusted-$p < 10^{-9}$), uterine corpus endometrial carcinoma (5.6-fold, $p < 10^{-8}$), prostate adenocarcinoma (3.7-fold, $p < 10^{-7}$), colon carcinoma (1.5-fold, $p < 10^{-3}$) etc[62]. Besides solid tumors, there is now evidence that cell-surface carbohydrates are altered during myelodysplastic neoplasms[63], and studies with sCore2 may add new dimensions to our understanding of such hematologic malignancies. The expanded use of sCore2 for such clinical investigations is part of our ongoing work.

In conclusion, we developed a sialoglycan-binding protein by engineering GTs, and this lectin displays glycan binding profiles different from other known lectins and antibodies. We also developed a platform technology for screening GT function. Extension of this approach could result in a rational strategy for generating new families of GBPs, based on the carbohydrate acceptor preference of the parent enzyme. This could also result in new GBPs for profiling healthy and disease human tissue, and related expansions to uncover the human glycome.

## Methods

### Ethical statement
Fresh human blood used in spectral flow studies was obtained from healthy non-smoking adult female and male volunteers (age > 18) by venipuncture. Written informed consent was obtained following protocols approved by the University at Buffalo Health Sciences Institutional Review Board (HSIRB protocol: CCE0011101E). The purpose of the research was explained to the donors, who were modestly reimbursed for their participation. Deidentified tissue sections described in this paper constitute non-human subject research.

## Materials
A 25-color immunoprofiling assay cFluor kit (Product code: R7-40002) was purchased from Cytek Biosciences (Fremont, CA). Monoclonal antibodies (mAbs) against human IgM, CD38, PD-1, CD14, CD16, and CD123 were removed from this panel, in order to accommodate additional fluorescent lectins and to expand the number of gated cell types. These mAbs were replaced by anti-CD14 (Clone: 63D3-BV711, Product code: 367139), anti-CD16 (Clone: 3G8-BV785, Product code: 302045), and anti-CD123 (Clone: 6H6-BV510, Product code: 306021) from BioLegend (San Diego, CA), and anti-CD15 (Clone: HI98-BUV395, Product code: 563872) from BD Biosciences (Franklin Lakes, NJ). Goat anti-human Fcγ fragment-specific IgG in unconjugated, Horseradish Peroxidase (HRP)-conjugated, AF488 (AlexaFluor 488)-conjugated, and AF647-conjugated forms (Product codes: 109-005-190, 109-035-098, 109-545-098, and 109-605-098, respectively) were from Jackson ImmunoResearch (West Grove, PA). Recombinant human ST3Gal1 (rhST3Gal1, Product code: 6905-GT-020), Recombinant human P-selectin Fc chimera (P-selectin-Fc, Product code: 137-PS-050), sheep anti-hGCNT1 IgG (Product code: AF7248-SP) and HRP donkey anti-sheep H + L IgG (Product code: HAF016) were from R&D Systems (Minneapolis, MN). Unconjugated Peanut Agglutinin (PNA, Product code: L-1070), *Maackia Amurensis* Lectin II (MALII, Product code: L-1260), *Erythrina Cristagalli* Lectin (ECL, Product code: L-1140), *Phaseolus Vulgaris* Leucoagglutinin (PHA-L, Product code: L-1110), and H.O.H (Human on Human) Immunodetection kit (Product code: HOH-3000) were from Vector Laboratories (Newark, CA). Galβ1,3GalNAcα-*p*NP (*para*-nitrophenol) was available from previous work[37]. CMP-Neu5Ac (cytidine 5′-monophospho β-D-*N*-acetylneuraminic acid) sodium salt (Product code: MC04391) was from Biosynth (Staad, Switzerland). α2-3, 6, 8, 9 Neuraminidase A (sialidase, Product code: P0722) was from New England Biolabs (Ipswich, MA). Sulfo-NHS (*N*-hydroxysuccinimide) (Product code: 24510), CellTracker Green CMFDA dye (Product code: C2925), and CellTracker Orange CMTMR dye (Product code: C2927) were from ThermoFisher (Waltham, MA). EDC (1-Ethyl-3-(3-dimethylaminopropyl)carbodiimide hydrochloride, Product code: BC25) was from G-biosciences (St. Louis, MO). AZDye 488 NHS (*N*-hydroxysuccinimidyl) ester (Product code: 1013) and AZDye 594 NHS ester (Product code: 1101) were from Fluoroprobe (Scottsdale, AZ). CF532 Dye SE/TFP Ester, CF594 Dye SE/TFP Ester, and CF700 Dye SE/TFP Ester (Product code: 92104, 92132, and 96067, respectively) were from Biotium (Fremont, CA). Hematoxylin solution (Mayer's, Modified, Product code: ab220365) was from Abcam (Waltham, MA). Permount mounting medium (Product code: 17986-05) was from Electron Microscopy Sciences (Hatfield, PA). All other chemicals were from Sigma, cell culture reagents from LifeTechnologies /ThermoFisher and molecular biology reagents from New England Biolabs, unless otherwise specified.

## Fluorescent labeling of lectins and antibodies
The buffer used for all lectins or antibodies (PNA, MALII, ECL, PHA-L, and goat anti-human Fcγ fragment-specific IgG) was exchanged to PBS (phosphate-buffered saline) using Zeba™ Spin desalting columns (7 K MWCO, 0.5 mL). For labeling, 5-fold molar excess AZDye or CF dye was added to lectins or Abs for 1 h at RT in the dark. Reaction was then quenched with 1/10 volume of 1 M Tris, followed by buffer exchange to PBS.

## Molecular Biology
Primers used in this study are listed in Supplementary Data 4. For constructing pCSCG-Fc-pST3Gal1 WT, human Fc and pST3Gal1 were amplified and connected via 5xGS linker sequence by overlap extension PCR. PCR product and pCSCG plasmid from previous study were digested with AgeI/BstBI, followed by gel purification and ligation[64]. For pCSCG-Fc-pST3Gal1 mutant (H302A, Dead, and individual mutants

from Lib2), pCSCG-Fc-CBM40 and pCSCG-Fc-diCBM40, PCR amplification was conducted using primers with 15-20 base pair overlaps and assembled using NEBuilder Hifi Assembly kit (New England Biolabs) or In-fusion Snap Assembly kit (Takara Bio). pET45b-CBM40 plasmid was kindly provided by Dr. Anne Imberty (University Grenoble Alpes, CNRS, CERMAV)[28]. For constructing surface display form of pCSCG-Fc-pST3Gal1, PCR amplification of cytoplasmic tail and transmembrane domain portion of human DPP4 was done using TEMP2 oligonucleotide as a template (Supplementary Data 4). The PCR amplicon was inserted into pCSGC-Fc-pST3Gal1 using shared NheI/XbaI sites. For constructing pCSCG-Fc-pST3Gal1 H302A mutant library, PCR amplification of vector backbone was done using 31MutaVec_FWD and 31MutaVec_REV (Supplementary Data 4). PCR amplification of mutagenic insert was done in 2 steps. In the 1st step, 31MutaIns_FWD and mutagenic reverse primers were used for Lib1 followed by gel extraction. Similarly, 31MutaIns_REV and mutagenic forward primers were used for Lib2. In the 2nd step, 31MutaIns_REV and megaprimers from the 1st step were used for Lib1 construction. Similarly, 31MutaIns_FWD and megaprimers from the 1st step were used for Lib2. After gel extraction of each mutagenic fragment for insert, 3 fragments for Lib1 and 5 fragments for Lib2 were pooled and subjected to assembly with vector backbone using Hifi Assembly kit. Products from Hifi assembly were electroporated into NEB 10-beta competent *E.coli* (New England Biolabs) following manufacturer's instructions. The number of colonies formed on the agar plate was checked so that there was at least 100x colony representation for each mutant in the library.

## Cell culture

HEK293T, HL60 and Calu-3 cells were from American Type Culture Collection (Manassas, VA). COLO357-FG cells were from Dr. Moorthy Ponnusamy (University of Nebraska Medical College). Human embryonic kidney 293T cells (HEK293T) were cultured in Dulbecco's Modified Eagle Medium (DMEM). HL-60 cells were cultured in Iscove's Modified Dulbecco's Medium (IMDM). COLO357-FG cells were cultured in Roswell Park Memorial Institute (RPMI) 1640 medium. Calu-3 cells were cultured in DMEM supplemented with 1% MEM non-essential amino acid solution. All media were supplemented with 10 % fetal bovine serum (FBS), 1 % Antibiotic-Antimycotic and 1 % GlutaMAX supplement. All cells were cultured at 37 °C in humidified 5 % $CO_2$ atmosphere.

## CRISPR-Cas9 isogenic clones

HEK293T lacking the human α(1-3)-mannosyl-glycoprotein 2-beta-*N*-acetylglucosaminyltransferase *MGAT1* (*MGAT1*-KO) and core-1 synthase *C1GalT1* (*C1GalT1*-KO) were previously established[29]. HL60 lacking *ST3Gal4* was described previously[42]. To make isogenic HEK293T clones lacking glucosaminyl (*N*-acetyl) transferase-1 *GCNT1* (*GCNT1*-KO), solute carrier family 35 member A1 *SLC35A1* (*SLC35A1*-KO), and ST3 β-galactose α(2-3)sialyltransferase-1 *ST3Gal1* (*ST3Gal1*-KO), single-guide RNAs (sgRNAs) targeting these genes were cloned into pX330-U6-Chimeric_BB-CBh-hSpCas9 vector (Addgene, plasmid #42230). Target sequences for each gene were: 5'-TGCTGAG-GACGTTGCTGCGA-3' and 5'-TCAGACACTTGGAGCTTGCT-3' for *GCNT1*, 5'-TTCTGTGATACACACGGCTG-3' and 5'-TGGGTATA-GACTGCAGCCATC-3' for *SLC35A1*, 5'-TGGAGGACGACACCTACCGA-3' and 5'-GAACTACTCCCACACCATGG-3' for *ST3Gal1*. A mixture of two plasmids containing sgRNAs targeting distinct sites on the target gene were pooled and transfected into HEK293T wildtype cells using the calcium phosphate method. Isogenic clones for each KO cell were obtained by single-cell FACS sorting. Fluorescently labeled MALII and PNA were used to assist sorting of *SLC35A1*-KO and *ST3Gal1*-KO cells. For *GCNT1*, single-cell sorting did not use any markers. After scale-up, genomic DNA was extracted from each cell lysate using PureLink™ Genomic DNA Mini Kit (Invitrogen). Target gene editing sites were PCR amplified, Nextera dual index adapters were appended, and then

150 bp paired-end amplicon sequencing (NGS) was performed using Illumina sequencers to confirm gene editing.

## Molecular surface area calculations

Sialoglycan ligand from 5FRE was overlayed into the original acceptor ligand of 2WNB to predict the sialic acid-binding pocket of pST3Gal1. Atoms within 4 Å from the ligand were defined as the binding interface between protein and glycan ligands. Crystal structures of all glycan-related proteins co-crystalized with naturally derived glycan ligands except for DANA, a transition state analog of sialic acid, were from the protein data bank. These were visualized using PyMOL. Molecular surface areas of defined binding interfaces were calculated by get_area command. Sampling density was set to 4.

## Fc-fusion protein expression and purification

Plasmids encoding for soluble Fc-fusion proteins were transiently transfected into HEK293T cells using the calcium phosphate method. To this end, cells were plated in 100 mm or 150 mm cell culture petri dishes. After overnight culture, medium was replaced with fresh DMEM 1 h before transfection. 1/10 cell culture volume of transfection mixture was prepared by mixing calcium chloride (final concentration; 125 mM) with plasmid DNA followed by addition of 2x HBS buffer (concentrations; 50 mM HEPES, 10 mM KCl, 140 mM NaCl, 1.5 mM $Na_2HPO_4$, pH 7.05). 25 or 55 μg plasmid DNA were used for 100 mm and 150 mm cell culture petri dishes, respectively. After treating cells with 25 μM chloroquine, transfection mixtures were added dropwise onto cells. Six h post-transfection, cell culture medium was replaced with serum-free DMEM lacking phenol red, but with 1 % MEM non-essential amino acid solution, 1 % Insulin-Transferrin-Selenium-ethanolamine, 0.4 g/L AlbuMAX™ Lipid-rich BSA, 1 % Antibiotic-Antimycotic and 1 % GlutaMAX supplement. Culture supernatant (10 mL for 100 mm dish and 20 mL for 150 mm dish) was collected 3 days post-transfection, cell debris was removed by centrifugation (3000 *g*, 3 min) and then the material was concentrated using Amicon Ultra centrifugal filter units (30 kDa MWCO) by 20 - 40-fold. Concentrated supernatant was dissolved in 10 mL PBS and mixed with 1 mL NEBExpress Ni resin (New England Biolabs) pre-equilibrated with 20 mL PBS. The mixture of resins and supernatant was incubated end-over-end at 4 °C for 15 min. Following centrifugation (800 *g*, 1 min), the resin was washed with 30 mL PBS containing 10 mM imidazole. Fc-fusion proteins were then eluted with 10 mL PBS containing 200 mM imidazole. The elution fraction was immediately concentrated using Amicon Ultra centrifugal filter units (30 kDa MWCO) and buffer-exchanged using Zeba™ Spin desalting columns (7 K MWCO, 0.5 mL) to PBS.

## Western blot

Cell culture supernatant was mixed with SDS loading buffer. 33 mM DTT was added for runs conducted under reducing conditions. All samples were denatured by heating at 95 °C for 5 min, loaded onto 4-12 % Tris-glycine SDS-PAGE gels, resolved and transferred onto nitrocellulose membranes. Following blocking with 5 % non-fat milk in TBST (20 mM Tris-HCl, 100 mM NaCl, 0.1 % Tween-20) for 1 h at RT, the membrane was incubated at 4 °C overnight in a TBST solution containing 1:5000 HRP-conjugated goat anti-human Fcγ fragment-specific IgG and 2 % non-fat milk. Following additional washes, signal was developed using SuperSignal™ West Pico PLUS Chemiluminescent substrate. Images were acquired using ChemiDoc Imaging system (Bio-Rad).

## Fc-protein quantitation using cytometry FLICA

A fluorescence linked immune-coupled assay (FLICA) was used to quantify Fc-protein concentration. To generate microspheres bearing anti-Fc Ab, 80 × 10^6 carboxylate microsphere beads (3 μm size, Product code: 09850, Polysciences, Warrington, PA) were washed twice

using PBS and dissolved into 300 μL MOPS buffer (20 mM MOPS, pH 6.0), followed by addition of 0.25 M EDC and 0.25 M sulfo-NHS (total reaction volume: 500 μL). After 30 min incubation at RT, the microspheres were washed five times using PBS, and supplemented with 150 μL goat anti-human Fcγ fragment-specific IgG (stock: 1.3 mg/mL) in 1 mL PBS. After 3 h at RT, the beads were centrifuged (14000 rpm, 6 min) and resuspended in PBS containing 40 mM ethanolamine at RT for 30 min. The final beads were washed and stored in 1 mL PBS containing 1% BSA at 4 °C prior to use. Efalizumab (humanized anti-CD11a, Genentech) was used as a standard for Fc-fusion protein quantification. To this end, Efalizumab serially diluted standards (6.25-200 ng/mL) or Fc-proteins were added to 2 μL FLICA beads in PBS containing 1 % BSA. Following 20 min incubation on ice, the beads were washed and resuspended into PBS containing 1 % BSA and 1:200 AF488-conjugated goat anti-human Fcγ fragment-specific IgG (3.75 μg/mL). The samples were placed on ice for 20 min, washed and analyzed using a flow cytometer. Calibration curves created using serial dilution of Efalizumab were used to determine Fc-fusion protein concentrations. FACSDiva 8.0.1 software was used for data analysis.

## LC-MS/MS enzymatic analysis
1 μL PBS (negative control), 1 μL commercial rhST3Gal1 (positive control), or 1.5 μg/mL PS1, H302A, or Dead were added to the mixture of 15 μL 100 mM cacodylate buffer (pH 6.0), 2 μL Galβ1,3GalNAcα-pNP (5 mM), and 2 μL CMP-Neu5Ac sodium salt (5 mM). Following overnight incubation at 37 °C, the reaction was quenched by addition of 80 μL acetonitrile to each sample at 4 °C for 30 min. These samples were then centrifuged at 4 °C (13000 g, 10 min), 90 μL of the supernatant was collected and then dried using Savant™ SPD131DDA SpeedVac Concentrator (Thermo). All samples were stored at -20 °C prior to LC-MS/MS analysis. Prior to injection, dried samples were dissolved in 50 % methanol (0.1 % formic acid). Each sample was separated on XSelect C18 column (3.5 μm, 4.6 mm × 150 mm) before being subjected to MS/MS analysis using Q-Exactive Hybrid Quadrupole-Orbitrap Mass Spectrometer (Thermo) in positive mode with HCD collision energy set to 28%. Flow rate was 0.2 mL/min and the column was maintained at 40 °C while running samples. Mobile phases were (A) MilliQ (0.1 % formic acid) and (B) acetonitrile (0.1 % formic acid). The gradient of liquid chromatography was as follows: (i) 0–40 % B (0–20 min), (ii) 40–100 % B (20–25 min), and (iii) 100–0 % B (25–30 min). Molecular weights used for obtaining XICs (extracted ion chromatogram) were as follows: donor substrate ($[M + H]^+ = 615.1551$), acceptor substrate ($[M + H]^+ = 505.1670$), and product ($[M + H]^+ = 796.2624$).

## Flow cytometry
All flow cytometry assays using live cells were performed in the mileu of HEPES buffer (30 mM HEPES, 110 mM NaCl, 10 mM KCl, 2 mM MgCl₂, 10 mM glucose, 1.5 mM CaCl₂, 1 % BSA, pH 7.4) with some of the steps being semi-automated using an OT-2 robot (Opentrons, Brooklyn, NY). In case of sialidase treatment, these cells in 200 μL volume were incubated with 2 μL sialidase (200 units/mL) or 2 μL PBS control (without sialidase) at 37 °C for 1 h, prior to resuspending in 200 μL HEPES buffer. Prior to cell staining, 2 μg/mL Fc-fusion proteins were pre-complexed with AF488- or AF647-conjugated goat anti-human Fcγ fragment-specific IgG (3 μg/mL) in HEPES buffer on ice for 10 min. During the labeling step, 5 μL cell suspension ($10^7$ cells/mL) was mixed with 5 μL fluorescently-labeled lectins (5 μg/mL) or pre-complexed Fc-fusion proteins (1 μg/mL) on ice for 20 min. Cell samples were diluted by addition of 90 μL HEPES buffer, washed and analyzed using a BD LSRFortessa™ X-20 flow cytometer (BD Biosciences). For binding assay using Fc-pST3Gal1 H302A mutants identified from Lib2, Fc-fusion proteins (final concentration: 5 μg/mL) and AF488-conjugated goat anti-human Fcγ fragment-specific IgG (final concentration: 7.5 μg/mL) were used.

In variations of the above protocol, in dose-dependence studies, Fc-fusion proteins were pre-complexed with AF488-conjugated anti-human Fcγ fragment-specific IgG at 2:3 ratio as above and added to cells, prior to fluorescence detection. In other cases, $4 × 10^5$ HL60 cells were seeded in 6-well plates prior to addition of 50 mM sodium chlorate in HEPES buffer for 2 days in standard tissue culture incubators. Cells were then resuspended at $10^7$ /mL, and their ability to bind 1 μg/mL of various Fc-fusion proteins (PS1, H302A, and P-selectin-Fc) was measured using the above method.

## Glycan microarray
Binding assay using Fc-proteins (PS1 and H302A) and glycan microarrays was performed following the protocol described at the NCFG website (Glycan binding assay with fusion or epitope tagged proteins:(https://research.bidmc.org/ncfg/protocols/glycan-binding-assay-fusion-or-epitope-tagged-protein). Briefly, hydrated CFG slides were incubated with 50 or 5 μg/mL PS1 or H302A in a humidified chamber for 1 h at RT. After washing, the slides were incubated with AF488-conjugated anti-human IgG Fc in a humidified chamber for 1 h at RT. Slides were then scanned to detect fluorescence following washing.

## CRISPR forward genetic screen
Both magnetic sorting and flow cytometry sorting were used for CRISPR screens. In the case of magnetic sorting, 500 μL M-450 Epoxy Dynabeads ($2 × 10^8$ beads, Product code: 14011) were washed three times with 1 mL PBS. 100 μg goat anti-human Fcγ fragment-specific IgG (200 μg/mL) was then added and incubated end-over-end overnight at RT. The reaction was then quenched using 25 mM ethanolamine for 30 min at RT. Following three additional washes using 1 mL PBS containing 1 % BSA, beads were stored at 4 °C until use. $5 × 10^6$ HL60 glycoCRISPR library cells, available from previous work[30], were suspended in HEPES buffer at $10^7$ cells/mL. 10 μg/mL Fc-fusion proteins were added to these cells for 20 min at 4 °C. Following washing with HEPES buffer, $5 × 10^6$ Dynabeads bearing anti-Fc Ab were added to the cells in 2.5 mL volume for 30 min at 4 °C. The cell-bead mixture was then separated by placing sample tubes in MagnaRack™ -magnetic separation racks (Invitrogen) for 2 min and resulting supernatant was collected. The collected cells were then washed and cultured in IMDM culture medium for scale-up and additional sorting. Three rounds of cell enrichments were conducted in this manner to obtain cell populations that did not bind Fc-proteins. In the case of cytometry sorting, sCore2 was pre-complexed with AF488-conjugated goat anti-human Fcγ fragment-specific IgG as described above for 10 min on ice. HL60 glycoCRISPR library cells (0.5 mL at $10^7$ cells/mL) were mixed with 1 μg/mL pre-complexed Fc-fusion proteins on ice for 20 min prior to sorting. In this case, two cycles of enrichment were performed to sort for non-binders. Genomic DNA was extracted from ~$5 × 10^6$ cells from the above preparation using the PureLink™ Genomic DNA Mini Kit. The sequences proximal to the sgRNA site were amplified as previously described[30,36], and this was subjected to NGS using either Illumina MiSeq or NovaSeq-X (~2 million reads/sample, 150 bp paired end). sgRNA enrichment analysis was performed using MAGeCK 0.5.9 package[65].

## Protein structural prediction
The AlphaFold model version, AF-Q02745-F1-v4, was used to predict the structure of pST3Gal1. pLDDT scores overlaid in the predicted structure and predicted aligned error (PAE) are shown in Supplementary Fig. 12.

## Transient surface-display of Fc-pST3Gal1 proteins
For surface display of Fc-fusion proteins, HEK293T cells plated in 6-well plates were transiently transfected using the above calcium phosphate method, only with 3 μg plasmid DNA/well. At 6 h, the

medium was replaced with DMEM with phenol red supplemented with 10 % FBS, 1 % Antibiotic-Antimycotic and 1 % GlutaMAX. Following overnight culture, cells were tripsinized and used for binding assay.

For cell-cell interaction assay for checking cis-interaction, HEK293T cells untransfected, mock transfected or transfected to display TM-PS1, TM-H302A, or TM-Dead were prepared. While some cells were sialidase treated, others served as PBS control. Here, the transfected cells (at $10^7$ cells/mL) were labeled with CellTracker Orange CMTMR dye (20 nM), and untransfected control HEK293T cells were labeled with CellTracker Green CMFDA dye (100 nM) by incubating for 30 min at 37 °C. After washing three times, 5 μL of green cells were mixed with 5 μL orange surface-displaying cells for 20 min at RT. Flow cytometer was then used to measure the formation of green-orange cell adhesion events.

### Fluorescent sialylated core-2 polyacrylamide (PAA) polymer preparation

GlcNAcβ1-6(Galβ1-3)GalNAcα-sp3-PAA-fluo (core 2-PAA-FP, MW - 30 kDa, Product code: 0078-FP) was purchased from GlycoNZ (Auckland, New Zealand). This polymer was dissolved in PBS at 1 mg/mL. 20 μL of this reagent was then mixed with 1 μL commercial rhST3Gal1 and 4 μL CMP-Neu5Ac sodium salt (final concentration: 800 μM) overnight at 37 °C. The reaction was stopped by removing excess unreacted CMP-Neu5Ac using Amicon Ultra centrifugal filter units (3 kDa MWCO). PAA-polymer concentration was determined by comparing the absorbance at 488 nm of the original Core-2-PAA-FP, with the reagent following sialylation.

### Surface display and selection of Fc-pST3Gal1 H302A mutant libraries

Lentivirus pools were produced for stable surface-display of Lib1 and Lib2 constructs using the pCSCG-Fc-pST3Gal1 H302A mutant libraries. To make virions, the plasmid libraries were transiently transfected into HEK293T cells using the calcium phosphate method (see above), along with packaging plasmid (psPAX2, Addgene_#12260) and envelope plasmid encoding for VSV-G (pMD2.G, Addgene_#12259). Transfections were performed in 150 mm cell culture petri dishes using 55 μg total plasmid, with the molar ratio of transfer (pCSCG), packaging, and envelope plasmids varying as 2:2:1. Six h post-transfection, the cell culture medium was replaced with 20 mL OptiMEM, with first virus batch being collected after overnight culture. Fresh 20 mL OptiMEM with 10 mM sodium butyrate was then added, with second virus batch was collected the next day. Both viral batches were then pooled, centrifuged (2000 g, 2 min) to remove debris, filtered using polyethersulfone (PES) syringe filters (0.45 μm), and subjected to ultracentrifugation (50,000 g, 2 h, 4 °C). The resulting lentivirus pellet was dissolved in 100 μL OptiMEM, aliquoted and stored at -80 °C.

For lentiviral transduction, HEK293T SLC35A1 KO cells were plated in 60 mm cell culture petri dishes. Cell culture medium was replaced with DMEM supplemented with 8 μg/mL polybrene for 5 min, and the lentivirus library pool was then serially diluted onto the cells. Cell culture medium was replaced to DMEM after overnight incubation. Following further overnight incubation, the trypsinized transduced cells were mixed with AF647-conjugated goat anti-human Fcγ fragment-specific IgG for 20 min on ice. Cell surface Fc expression was measured using flow cytometry as a surrogate measure of % viral transduction. The viral titer resulting in 20–30 % Fc-positive cells was scaled up, and FACS sorted to constitute the mutant libraries.

Mutants in Lib1 and Lib2 displaying enhanced binding to sialyl core-2-PAA-FP were sorted and sequenced. To this end, $10^7$ library cells/mL suspended in HEPES buffer were mixed with sialyl core-2-PAA-FP (5 μg/mL) and AF647-conjugated goat anti-human Fcγ fragment-specific IgG (0.075 μg/mL) for 20 min on ice in ~500 μL volume. AF647-positive cells with high sialyl core-2 PAA-FP binding were sorted. Following scale up, genomic DNA was extracted from $5 \times 10^6$ unsorted or

FACS sorted HEK293T Lib1 and Lib2 cells. The region spanning the pST3Gal1 mutations was PCR amplified (primers in Supplementary Data 4), barcoded and subjected to NGS using Illumina MiSeq Micro kit (4 million reads, 150 bp paired end). Mutant enrichment analysis was performed using Seqkit[66]. First, the read number of each mutant was divided by the total read number to calculate the read number ratio. Negative, 1st positive and 2nd positive enrichment values were then calculated by dividing the read number ratio of these samples by the read number ratio of sample before sorting. Enrichment scores were calculated by multiplying 1st positive and 2nd positive enrichment values and dividing it by negative enrichment value.

### Spectral flow cytometry

5 mM EDTA was used as anti-coagulant for human blood draws. Buffy coat was obtained from this blood by centrifugation (1000 g, 12 min). Red blood cells (RBC) were lysed from buffy coat by addition of 30 mL RBC lysis buffer (10 mM KHCO₃, 155 mM NH₄Cl, 1 mM Na₂EDTA) for 10 min at RT, followed by washing with 5 mL HEPES buffer. The resulting white blood cells were resuspended at $25 \times 10^6$ cells/mL, and incubated with the 23 anti-human leukocyte fluorescent mAb panel. $1.5 \times 10^6$ cells were labeled in this manner for 30 min at 4 °C in 125 μL volume, both for the mixed panel and single-stain Ab controls. Labeled cells were washed using HEPES buffer and fixed overnight using 0.5 % paraformaldehyde at 4 °C. The following day, the cells were again washed using HEPES buffer, and incubated with 1 μg/mL CF532-PNA, CF594-MALII, and Fc-proteins pre-complexed with 1.5 μg/mL CF700-goat anti-human Fcγ fragment-specific IgG for 30 min at 4 °C. Pre-complex was formed by incubating for 10 min at 4 °C prior to cell labeling. Then, washing using HEPES buffer and data acquisition on Cytek Aurora spectral flow cytometer were performed. Unmixing was done by SpectroFlo v3.2.1 software and compensation correction using FlowJo (BD Biosciences). tSNE plots were created using Bokeh (bohek.org). For heatmaps, the medians for each population of lectins were scaled from 0 to 1 before plotting the results.

### Tissue microarray analysis

Paraffin-embedded deidentified human tissue microarrays were kindly provided by the Cooperative Human Tissue Network (CHTN) at the University of Virginia (CHTN_Norm3 and CHTN_Test3). All tissues were deparaffinized using xylene (1 × 10 min) and rehydrated with 100 %, 95 %, 70 % and 50 % EtOH (1 × 10 min for 100 % EtOH and 1 × 2 min for others) and MilliQ water (1 × 1 min). Following antigen retrieval in 10 mM citrate buffer (pH 6.0) at 96 °C for 45 min and washing with PBS (2 × 5 min), endogenous peroxidase activity was blocked using 3 % hydrogen peroxide at RT for 10 min and washed with PBS (2 × 5 min). Tissue sections were treated with sialidase (200 U/mL) in PBS (0.1 % BSA) at 37 °C for 1 h. The tissue samples were then washed with PBS (2 × 5 min), before being blocked with protein block solution provided with the H.O.H immunodetection kit. Tissue sections were then incubated with either 20 μg/mL sCore2 or Dead pre-complexed with 15 μg/mL HRP-goat anti-human Fcγ fragment-specific IgG overnight at 4 °C. Pre-complexing was performed by incubating sCore2 or Dead mixed with HRP-goat anti-human Fcγ fragment-specific IgG in PBS (0.1 % BSA) at RT for 30 min, adding solution-B from H.O.H immunodetection kit into the mixture, and then further incubating at RT for 30 min. Combination of pre-complex method and human-on-human detection kit reduces human Fc-derived background for human tissue[67]. For staining using anti-GCNT1 antibody, tissues were incubated with 3 μg/mL sheep anti-GCNT1 overnight at 4 °C, followed by washing with PBS (2 × 5 min) and incubation with 5 μg/mL HRP-donkey anti-sheep for 30 min at room temperature on the next day. Following washing with PBS (2 × 5 min), tissues were incubated with ImmPACT DAB (3,3'-diaminobenzidine) EqV Reagents in immunodetection kit for 5 min to develop brown-colored precipitates. These sections were then washed with PBS (2 × 5 min), counterstained using 50 % hematoxylin diluted

with MilliQ, dehydrated with 95 % EtOH, 100 % EtOH and xylene (1 × 1 min for each) and covered using permount mounting medium. Whole slides were scanned using Aperio Versa 200 (Leica Biosystems).

## Statistical analysis

All data are presented as mean ± standard deviation (STD). Dual comparisons were performed using two-tailed unpaired Student's $t$-test. Multiple comparisons were performed using one-way ANOVA followed by the Tukey post-test. $p < 0.05$ was considered to be statistically significant. Number of repeats are specified in individual panels.

## Reporting summary

Further information on research design is available in the Nature Portfolio Reporting Summary linked to this article.

## Data availability

All data supporting the results of this study can be found in the article, supplementary, and source data files provided with this article. Source Data are also available at Figshare (https://doi.org/10.6084/m9.figshare.28909301). NGS data are deposited as part of NCBI BioProject: PRJNA1173749. Raw LC-MS data are deposited at Zenodo (https://zenodo.org/records/15793727). Plasmids for Fc-pST3Gal1 WT, H302A, Dead and sCore2 are deposited at Addgene. Any other plasmids described in this paper will be provided by the corresponding author. Source data are provided with this paper.

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

## Acknowledgements

This work was supported by National Institutes of Health grants GM139160, HL103411 and HL151333. The Next-generation sequencing and Flow and Image Cytometry Shared Resources (FICSR) at the Roswell Park Comprehensive Cancer Center (RPCCC) were partially supported by NCI grants P30CA01656 and NCI R50 R50CA211108. Glycan microarray assays performed at the National Center for Functional Glycomics (NCFG) at the Beth Israel Deaconess Medical Center, Harvard Medical School, were supported by R24 GM137763. We are grateful to Jamie Heimburg-Molinaro and Akul Mehta for performing these assays. Tissue microarrays were kindly provided by the NCI Cooperative Human Tissue Network (CHTN). Some figures in this paper were created with BioRender.com.

## Author contributions

R.H.: Investigation (all experiments), Data Curation, Writing–Original Draft, Formal Analysis. L.E.B.: Investigation (spectral flow cytometry studies). J.T.: Investigation (tissue microarray analysis and scoring). S.P.: Conceptualization, Supervision, Formal Analysis. S.N.: Conceptualization, Supervision, Data Curation, Writing – Review & Editing, Formal Analysis.

## Competing interests

A provisional patent application has been filed by the State University of New York, Buffalo, NY on behalf of R.H. and S.N. (US 63/684,349, 2024). The patent relates to the composition of glycan binding proteins described in this manuscript and methods to develop them. All other authors declare no competing financial and non-financial interests.
