## [Transparent Peer Review file · Nature Communications]

Engineering glycosyltransferases into glycan binding proteins using a mammalian surface display platform

Corresponding Author: Professor Sriram Neelamegham

Version 0:

Reviewer comments:

Reviewer #1

(Remarks to the Author)

This manuscript describes a novel method for detection of cell surface glycans, specifically binding to sialylated core-2 O-linked glycans. Initially a single point mutation was observed to significantly increase the binding of FcSTSGal1 but subsequent mutation screening revealed a substantial increase in binding affinity with additional point mutations. Interestingly, investigation of the sCore2 tool revealed differential O-linked glycosylation profiles in different cell types and potentially disease states. Therefore, this technology may be used as a novel screening method for the detection of different diseases, including cancer. This is a novel, well defined (binding specificity) tool with great potential.

The manuscript is well written and data convincing. I have few comments:

Pg 5 The authors comment that the PS1 and mutants did not express well in HEK293T cells but expression did not improve in C1GalT1-KO cells. They suggest that the protein might be sequestered in the secretory pathway intracellularly and not secreted. However, they did not show the relative amount of protein secretion compared to retained within the cell. Western blotting, flow cytometry or microscopy could have been used to determine if the protein is retained. If it is retained, is this inducing cellular stress. Does this impact protein production? Have the authors tried to express mutants in SLC35A1 or ST3Gal1 KO cells after defining the binding specificity? Would they predict that either of these would improve secretion?

This is important from the point of view that if this is to be used as a tool, the production of the sCore2 protein must be scalable. What is the yield?

Pg 7 Of the 562 glycans included in the microarray, how many are physiologically relevant/abundant?

Pg 10 The mutant screening of PS1 variants yielded high affinity binders for sialylated core-2 O-linked glycans, how could this technology be used to screen for high affinity binders of other glycans? Include more in the discussion.

In general, it is difficult to ascertain the binding affinity to get a feel for how much protein would be required for diagnostics etc. perhaps the authors could discuss or propose further binding studies.

Reviewer #2

(Remarks to the Author)

In the submitted manuscript, Hombu *et al.* engineer a glycan-binding protein by catalytic inactivation of a glycosyltransferase using site-directed mutagenesis. The authors note that this is the first time a glycan-binding protein or lectin has been engineered from lab directed mutagenesis of a glycosyltransferase. Such designed glycan-binding proteins can be very important tools for detection of specific glycan epitopes or biomarkers, having some valuable applications in medicine. The

authors demonstrate the specificity of the mutant glycan-binding proteins is complementary to existing and currently available glycan-binding proteins and they demonstrate binding to defined glycan structures on a microarray and also to a variety of cell types known to have distinct glycosylation patterns. The authors also developed a mammalian cell-surface display platform to screen mutant libraries for improved variants of the glycan-binding proteins. The work is important and well executed. The authors, however, can improve the paper by better placing the technology they developed within the context of the current state of the field.

While this appears to be the first example of a glycosyltransferase being engineered in the lab to generate a glycan-binding protein (as far as this reviewer is aware), the concept of converting carbohydrate-active enzymes (CAZymes) into glycan-binding proteins by inactivating catalytic residues is not new. Inactivation of glycoside hydrolases to generate glycan-binding proteins is part of a technology that has been commercialized as "Lectenz®" and glycan-binding proteins have also been generated from inactivating carbohydrate esterases. This is also observed in nature, where wild-type CAZymes evolve into "pseudoenzymes" that can still bind glycans but cannot catalytically turn them over. This has been well reviewed (*e.g.*, <https://doi.org/10.3390/molecules26020380>).

The mammalian cell-surface display seems to be a very interesting development. The authors did not cite any reference for mammalian cell-surface display—is this a new innovation? It would be very important for the authors to state this explicitly to help define the novelty of this work.

Some minor comments:

Page 3, paragraph 1: Glycosylation is more than just a post-translational modification, as macromolecules (and small molecules) other than proteins can be glycosylated.

Page 3, paragraph 2: The authors mention "degenerate" terminal epitopes. I'm not sure that this terminology fits here

Page 4, paragraph 4: The authors state "this strategy may be applied to other enzymes from diverse species"—this strategy (inactivating CAZymes by mutagenesis) has been applied to other enzymes... do the authors mean that it may be applied to other glycosyltransferases?

Page 7, paragraph 2 and 3: The use of numbers in square brackets was confusing as it usually denotes a citation

Page 8: "Unbound HL60 cells lacking glycan epitopes required for Fc-protein were negative-selected"—I'm not sure about the semantics here. Normally negative-selection means that the cells lacking some marker survive or are kept. Here they are discarded, whereas the HL60 cells that have glycans that bind to the glycan-binding protein are kept. To me it seems like a clear example of positive selection (of the cells that bind).

Page 8: "HL60 cells enriched with H302A showed a distinct negative population as its initial binding was strong and the decrease in lectin binding caused a population shift"—I'm not sure I understand this explanation. Is there some difference between the conditions of the first sort and the third sort?

Page 12: Discussion could provide more context on other lab-designed/engineered lectins.

Reviewer #3

(Remarks to the Author)

The paper by Hombu et al describes both the development of an exciting new reagent to study glycosylation and a pathway for future development of such reagents. Overall the paper is well written and presented. However, there are a few moderate issues that the reviewer would like to see addressed.

1. Although sCore2 appears to have a higher affinity binding than H302A, it is not clear that it is more specific as is intimated in the paper. Glycan microarray analysis and/or analysis using the HL-60 library to confirm specificity would aid in making this statement.

2. No sialyltransferase is seen in the H302A Cas9/CRISPR library. This is strange given the data on individual knockouts that clearly shows that ST3GAL1 abrogates binding. The discussion where it is intimated that H302A is more specific than the initial transferase for sialylation hinges on the idea that since ST3GAL4 didn't show up in the data, that means there is higher specificity. However, ST3GAL1 didn't show up. It would be appropriate to knockout ST3GAL4 and show that only PS1 binds it prior to making this claim. In addition, how well is the activity of ST3GAL4 characterized? Is it possible that there is some redundancy in making the same epitope as ST3GAL4? The better argument for PS1 specificity comes from the lack of non-sialylated binding observed for H302A.

3. What are the K_d's of the 2 reagents (or apparent K_d's). No characterization of the binding constants are given. Better characterization of this would be beneficial.

4. On a practical note, some mention of how easy, or hard, it is to make these reagents would be good. What is the average yield per x of cells? Is there a possibility of bacterial production or yeast production of these in the future? This is often a limitation for reagents to be useful. Although bacterial and yeast production are mentioned in terms of the mutant libraries, a

more pertinent point would be the potential for making these reagents. In addition, how stable are these reagents? This is often an issue with mammalian lectins and is one of the reasons that the field has focused on bacterial lectins and glycosidases for engineering.

5. Although the staining data is interesting, there is no discussion of whether this matches expectations for sCore2. Would it be possible to compare this to GCNT1 staining on at least a few tissues to see whether this would line up (which based on arguments made would be somewhat expected)?

Minor issues:

1. On p. 6 diCBM40 is described as having "weaker sialic acid dependence". The choice of words is inaccurate. The data clearly shows that the lectin is dependent on sialic acid. It seems more clear that there is lower binding affinity as might be expected for a carbohydrate binding module.

2. On p. 7 "these lectins did not also bind additional sialoglycans: should read: these lectins also did not bind.

Version 1:

Reviewer comments:

Reviewer #1

(Remarks to the Author)

I believe that the authors have provided satisfactory answers to my queries. The manuscript has been amended with additional data and written sections to improve the overall clarity and context. I have no new comments.

Reviewer #2

(Remarks to the Author)

The authors have addressed all of my comments.

Reviewer #3

(Remarks to the Author)

The authors have done a nice job answering the review. I support publication of the revised work.

Response to reviewer comments

We thank all reviewers for carefully reading our submission and providing constructive comments. Our response is provided in the revised manuscript and in this document using red fonts. Changes made to the revised manuscript are noted in the text below using **bold fonts**.

Reviewer #1

This manuscript describes a novel method for detection of cell surface glycans, specifically binding to sialylated core-2 O-linked glycans. Initially a single point mutation was observed to significantly increase the binding of FcSTSGal1 but subsequent mutation screening revealed a substantial increase in binding affinity with additional point mutations. Interestingly, investigation of the sCore2 tool revealed differential O-linked glycosylation profiles in different cell types and potentially disease states. Therefore, this technology may be used as a novel screening method for the detection of different diseases, including cancer. This is a novel, well defined (binding specificity) tool with great potential.

The manuscript is well written and data convincing. I have few comments:

Response: We thank the reviewer for kind words and insightful comments.

Pg 5 The authors comment that the PS1 and mutants did not express well in HEK293T cells but expression did not improve in C1GalT1-KO cells. They suggest that the protein might be sequestered in the secretory pathway intracellularly and not secreted. However, they did not show the relative amount of protein secretion compared to retained within the cell. Western blotting, flow cytometry or microscopy could have been used to determine if the protein is retained. If it is retained, is this inducing cellular stress. Does this impact protein production? Have the authors tried to express mutants in SLC35A1 or ST3Gal1 KO cells after defining the binding specificity? Would they predict that either of these would improve secretion?

This is important from the point of view that if this is to be used as a tool, the production of the sCore2 protein must be scalable. What is the yield?

Response: As PS1 and additional mutants have a secretion signal peptide they are not retained in cells. We are sorry that the Reviewer left with the impression that the protein was not well formed and sequestered in the ER, potentially resulting in unfolded protein response. To the contrary, the protein expressed well and was secreted at ~2.5-10µg/mL in culture supernatant. These levels are comparable to concentrations of other secreted proteins we have produced previously (example ref. 1). In each case, proteins were produced using identical calcium phosphate method to transfect substrate adherent HEK293T cells. **This is clarified in the revised manuscript.**

It could be that the studies presented using C1GalT1-KO contributed to the concern raised by the reviewer (Fig. S3a). In this regard, we only performed this study to test if the produced Fc fusion-lectins are sequestered on surface sialoglycans (not inside cells), and if preventing this binding may increase lectin concentrations in culture supernatant. To this end, we compared protein expression in wild-type (WT) cells versus C1GalT1-KO. Here, we observed that lectin concentrations in supernatant of C1GalT1-KO were not higher than levels seen in wild-type cells, suggesting that lectin binding to cell surface sialoglycans has a minimal effect on supernatant concentrations.

Based on the reviewer's suggestion, we now also tested the expression of 4 different Fc-fusion proteins (PS1, H302A, Dead, and sCore2) in SLC35A1-KO cells (**new Supplemental Fig. S3a**). As seen, protein concentrations in culture media is no different in these KO cells compared to unmodified wildtype cells. Again, only a small portion of the Fc-fusion protein is retained on the cell surface, and a vast majority is in suspension.

With respect to future/ongoing scale-up plans, in one aspect, we attempted bacterial expression by including our lectins as maltose-binding protein (MBP)-fusion constructs. Though the protein was expressed, it was unstable following MBP removal. This is consistent with previous studies that suggest challenges with expression of mammalian glycosyltransferases in bacteria ². Currently, we are scaling up these lectins using standard yeast (*S. cerevisiae*) in anticipation that this will be followed by protein expression in yeast och1 mutant cells in the near future. The latter cells would bear glycan binding proteins with shorter, human-like mannosylated glycans. We expect this work to enable large-scale production for this reagent and also establish a platform-technology for the scaleup of additional reagents currently in the pipeline.

Overall, several edits have been made to Results (in first section) and a new Supplemental Fig. S3a has been added to address this comment.

Pg 7 Of the 562 glycans included in the microarray, how many are physiologically relevant/abundant?

Response: We have individually examined the glycans in the microarray (see revised Table S1) and highlighted the physiological entities and partial epitopes in green based our knowledge of enzymology/ biochemistry. Based on this, ~80% of the 562 glycans in the microarray are physiological. We however note that new glycans and glycosylation pathways continue to be discovered. Thus, our estimate is approximate and not precise.

Pg 10 The mutant screening of PS1 variants yielded high affinity binders for sialylated core-2 O-linked glycans, how could this technology be used to screen for high affinity binders of other glycans? Include more in the discussion.

Response: Indeed, this is a great point. In principle, the technology we introduce can be used to modify any glycosyltransferase to bind either the acceptor, product or related entities. Fine tuning of binding specificity is possible using modern protein engineering, and this can be guided by machine learning. Additionally, our approach aims to develop lectins by design based on the binding specificity and activity of the starting glycosyltransferase/enzyme. This is unlike more conventional methods that attempt to obtain glycan binding proteins using screening methods. As suggested, **we provide more details on how this technology may be extended in the revised Introduction and Discussion.**

In general, it is difficult to ascertain the binding affinity to get a feel for how much protein would be required for diagnostics etc. perhaps the authors could discuss or propose further binding studies.

Response: In typical assays, we use 1-5µg/mL of the lectin for single-cell binding studies, which is comparable to traditional lectins. This concentration is sufficient to observe >10-fold higher signal for positive control cells compared to sialidase-treated samples. Typically, we purify ~100µg of the lectin fusion-protein after transfecting a single 150mm HEK cell dish, and this is sufficient for a large number of assays. For blood cell profiling we used the GBPs at 1µg/mL. For tissue microarray, the concentration was higher at 20µg/mL due to lower sensitivity of detection. **These relevant concentration values and product yield data are provided in the revised Results and Methods.**

Reviewer #2

In the submitted manuscript, Hombu *et al.* engineer a glycan-binding protein by catalytic inactivation of a glycosyltransferase using site-directed mutagenesis. The authors note that this is the first time a glycan-binding protein or lectin has been engineered from lab directed mutagenesis of a glycosyltransferase. Such designed glycan-binding proteins can be very important tools for detection of specific glycan epitopes or biomarkers, having some valuable applications in medicine. The authors demonstrate the specificity of the mutant glycan-binding proteins is complementary to existing and currently available glycan-binding proteins and they demonstrate binding to defined glycan structures on a microarray and also to a variety of cell types known to have distinct glycosylation patterns. The authors also developed a mammalian cell-surface display platform to screen mutant libraries for improved variants of the glycan-binding proteins. The work is important and well executed. The authors, however, can improve the paper by better placing the technology they developed within the context of the current state of the field.

Response: Thank you for your insightful comments. We have considered your suggestions carefully and edited the manuscript as appropriate.

While this appears to be the first example of a glycosyltransferase being engineered in the lab to generate a glycan-binding protein (as far as this reviewer is aware), the concept of converting carbohydrate-active enzymes (CAZymes) into glycan-binding proteins by inactivating catalytic residues is not new. Inactivation of glycoside hydrolases to generate glycan-binding proteins is part of a technology that has been commercialized as “Lectenz[®]” and glycan-binding proteins have also been generated from inactivating carbohydrate esterases. This is also observed in nature, where wild-type CAZymes evolve into “pseudoenzymes” that can still bind glycans but cannot catalytically turn them over. This has been well reviewed (e.g., <https://doi.org/10.3390/molecules26020380>).

Response: Thank you for raising these critical points. We had considered these previously in Introduction, and have now expanded this section to include the citation suggested by the reviewer³ and also two other citations related to the Lectenz approach⁴ and a more recent publication that uses a variant of this method to create sialic acid binding proteins⁵. **Please see revised Introduction for changes.**

The mammalian cell-surface display seems to be a very interesting development. The authors did not cite any reference for mammalian cell-surface display—is this a new innovation? It would be very important for the authors to state this explicitly to help define the novelty of this work.

Response: We have added text to the revised Discussion presenting examples of mammalian surface display. Indeed, they are less common compared to bacterial and yeast surface display in the protein engineering field, but they are still extensively used to display Chimeric Antigen Receptors (CARs) on immune cells⁶. They have also been used previously to screen for antibodies^{7,8}, viral protein mutations⁹ and engineer leguminous lectins¹⁰. While there is merit in each of these applications, most mammalian surface-display methods focus on the display of type-I transmembrane proteins, while the engineering of glycosyltransferases requires type-II transmembrane display. Searching through literature, we only found one example of type-II transmembrane display and that too for screening short cysteine-rich peptides¹¹. To our best knowledge, this manuscript is the first example of Type-II transmembrane surface display for engineering glycosyltransferases. **We explicitly state this in the revised Discussion.**

Some minor comments:

Page 3, paragraph 1: Glycosylation is more than just a post-translational modification, as

macromolecules (and small molecules) other than proteins can be glycosylated.

Response: We rephrased glycosylation as a “molecular modification”

Page 3, paragraph 2: The authors mention “degenerate” terminal epitopes. I’m not sure that this terminology fits here

Response: We rephrased this to “many related” terminal epitopes.

Page 4, paragraph 4: The authors state “this strategy may be applied to other enzymes from diverse species”—this strategy (inactivating CAZymes by mutagenesis) has been applied to other enzymes... do the authors mean that it may be applied to other glycosyltransferases?

Response: Yes, this can be applied to other glycosyltransferases. Based on your comment, we rephrased this part to “glycosyltransferases” so that there is less ambiguity.

Page 7, paragraph 2 and 3: The use of numbers in square brackets was confusing as it usually denotes a citation

Response: We replaced square brackets with round brackets to avoid confusion. In Nature family journals, citations appear as superscript.

Page 8: “Unbound HL60 cells lacking glycan epitopes required for Fc-protein were negative-selected”—I’m not sure about the semantics here. Normally negative-selection means that the cells lacking some marker survive or are kept. Here they are discarded, whereas the HL60 cells that have glycans that bind to the glycan-binding protein are kept. To me it seems like a clear example of positive selection (of the cells that bind).

Response: The reviewer may have misunderstood. In this assay, we sorted cells from the HL60 glycoGene-CRISPR library that did not bind the GBP. These unbound cells lack the epitope necessary for binding the Fc-fusion proteins (red histogram in Fig. 4b, compared to black histogram for unsorted/starting cells). As GBP binding to HL60s was reduced following sorting, this is termed “negative selection”. **The text in the main manuscript has been rewritten to enhance clarity.**

Page 8: “HL60 cells enriched with H302A showed a distinct negative population as its initial binding was strong and the decrease in lectin binding caused a population shift”—I’m not sure I understand this explanation. Is there some difference between the conditions of the first sort and the third sort?

Response: We apologize for the confusion and have fully revised this section for clarity. The only point being made here was that the negative-selected H302A library showed a distinct shift to the left in the cytometry panel compared to unsorted cell control (Fig. 4b, left panel), while this was less obvious for PS1 (Fig. 4c, left panel). This is likely because H302A binds with greater affinity and specificity compared to PS1. There was no difference in the sorting or other experimental conditions used in both assays.

Page 12: Discussion could provide more context on other lab-designed/engineered lectins.

Response: We feel we have discussed this extensively in Introduction. In response to the reviewer, we have edited this section to provide more context including citations to recent articles. **We have also added a few more citations in Discussion.**

Reviewer #3

The paper by Hombu et al describes both the development of an exciting new reagent to study glycosylation and a pathway for future development of such reagents. Overall the paper is well written and presented. However, there are a few moderate issues that the reviewer would like to see addressed.

Response: Thank you for taking time to review our work. We have addressed the comments by performing additional experiments and making text changes to clarify selected aspects.

1. Although sCore2 appears to have a higher affinity binding than H302A, it is not clear that it is more specific as is intimated in the paper. Glycan microarray analysis and/or analysis using the HL-60 library to confirm specificity would aid in making this statement.

Response: We have attempted both experiments suggested by the reviewer. Unfortunately, the glycan microarray studies were unsuccessful due to technical complications at the NCFG core for reasons unrelated to this project.

However, we successfully completed a CRISPR screen of sCore 2 using the HL60 knockout cell library. These studies used two rounds of flow cytometry sorting to negative-select for sCore2 non-binders followed by short-read Illumina sequencing (**details in revised Methods**). Sequencing after the first round resulted in several of the anticipated hits (left side of Table R1 below). The more penetrating hits were selected in the second round (right side of Table R1).

First cytometry sort				Second cytometry sort			
id	score	p-value	fdr	id	score	p-value	fdr
GNE	2.687×10^{-10}	4.966×10^{-6}	0.00043	C1GALT1C1	3.69×10^{-8}	4.966×10^{-6}	0.00025
SLC35A1	3.589×10^{-9}	4.966×10^{-6}	0.00043	CMAS	4.167×10^{-8}	4.966×10^{-6}	0.00025
NANS	3.807×10^{-9}	4.966×10^{-6}	0.00043	GNE	9.214×10^{-9}	4.966×10^{-6}	0.00025
ST3GAL4	3.908×10^{-12}	4.966×10^{-6}	0.00043	NANS	4.784×10^{-13}	4.966×10^{-6}	0.00025
ST3GAL1	7.24×10^{-6}	1.49×10^{-5}	0.00103	ST3GAL4	1.213×10^{-9}	4.966×10^{-6}	0.00025
C1GALT1C1	7.522×10^{-6}	2.483×10^{-5}	0.00143	GCNT1	1.882×10^{-7}	4.966×10^{-6}	0.00025
ALG9	7.464×10^{-6}	3.476×10^{-5}	0.00172	SLC35A1	5.459×10^{-11}	4.966×10^{-6}	0.00025
SLC35B2	7.524×10^{-5}	5.115×10^{-4}	0.02212	C1GALT1	1.417×10^{-6}	1.49×10^{-5}	0.00064
CMAS	1.357×10^{-4}	6.803×10^{-4}	0.02616				
C1GALT1	4.809×10^{-4}	2.627×10^{-3}	0.0908				

* MAGeCK analysis results. All hits with FDR<0.1 are shown.

The flow cytometry plot in the **new Fig.6e** shows that sCore2 binding was low in the sorted cells compared to the wild-type population. As seen, sgRNA enriched in both sorts contained genes related to sialic acid (*CMAS*, *GNE*, *NANS*, *SLC35A1*) and core-1 O-glycan (*C1GalT1C1*, *C1GalT1*) biosynthesis, confirming specific recognition of O-linked glycans. The core-2 O-glycan biosynthesis gene (*GCNT1*) was slightly below the fdr cut-off in the first cycle (fdr=0.046) but was prominently enriched after the second sort. Both *ST3Gal1* and *ST3Gal4* were noted in the samples from first sort, but only *ST3Gal4* is seen after the second sort.

Based on our previous work, it seems possible that in HL60 cells, *ST3Gal4*, in addition to *ST3Gal1*, may act on Type-III substrates ($\text{Gal}(\beta 1-3)\text{GalNAc}\alpha$). In this regard, though $\text{Gal}(\beta 1-4)\text{GlcNAc}\beta$ is the more prominent and commonly regarded substrate for *ST3Gal4*, this enzyme also acts on $\text{Gal}(\beta 1-3)\text{GalNAc}\alpha$. This is noted in our previous enzymology studies that compared K_M values for different human *ST3Gals*¹². Here the K_M for $\alpha(2-3)$ sialylation of $\text{Gal}(\beta 1-3)\text{GalNAc}\alpha$ -O-Benzyl substrate

varied as: ST3Gal1 ($K_M = 0.05 \text{ mM}$) > ST3Gal2 ($K_M = 0.13 \text{ mM}$) ~ST3Gal4 ($K_M = 0.14 \text{ mM}$)¹². Similar results are also reported by others¹³. Our MALDI-TOF/TOF analysis of O-glycans in HL60 cells also shows partial reduced sialylation on the core-1 O-arm upon knocking out ST3Gal4 (ref.¹⁴, Figure reproduced below). Thus, ST3Gal1 and ST3Gal4 may exhibit redundancy by contributing to sialyl core-2 O-glycan biosynthesis, with neither being penetrating. Due to this, these genes are sometimes missed in the CRISPR screens. However, only sialylation on the core1 arm results in GBP binding as seen in the glycan microarray results.

[Figure Redacted]

Supplemental Fig 4 of ref.¹⁴. MALDI-TOF MS spectra of permethylated O-glycans derived from HL-60 cells. MALDI-TOF MS spectra are from A. WT, B. ST3Gal-4-KO HL-60 cells.

To elaborate on this concept, we additionally tested the ability of sCore2 and additional lectins to bind HL60-KO cells lacking ST3Gal4¹⁴ (**new Fig. 6f**). Partial reduction in both H302A and sCore2 is observed, but such binding is abolished for PS1. **We have revised Results related to Fig. 4 and Fig. 6 to present the above findings.** This does not affect the overall conclusions of the manuscript.

2. No sialyltransferase is seen in the H302A Cas9/CRISPR library. This is strange given the data on individual knockouts that clearly shows that ST3GAL1 abrogates binding. The discussion where it is intimated that H302A is more specific than the initial transferase for sialylation hinges on the idea that since ST3GAL4 didn't show up in the data, that means there is higher specificity. However, ST3GAL1 didn't show up. It would be appropriate to knockout ST3GAL4 and show that only PS1 binds it prior to making this claim. In addition, how well is the activity of ST3GAL4 characterized? Is it possible that there is some redundancy in making the same epitope as ST3GAL4? The better argument for PS1 specificity comes from the lack of non-sialylated binding observed for H302A. **Response:** We agree with the comments of the reviewer and believe we have addressed this in response to the previous comment. It is indeed possible that there is some redundancy in ST3Gal1 and ST3Gal4, at least in HL60 cells where the glycoGene knockout library is made. Thus, the binding of GBP formed by engineering ST3Gal1 are dependent on the enzyme activity of both ST3Gal1 and

ST3Gal4. In addition to Fig. 4b of main manuscript (which presents NGS data after 3 rounds of sorting), such dependence is also seen after one-round of sorting H302A. These data suggest that both ST3Gal4 and ST3Gal1 contribute partially to forming the sialyl core-2 O-glycan, but these genes are excluded after the third sort due to incomplete penetration. **Edits have been made to text related to Figure 4 and Figure 6 to address this reviewer comment, and additionally new data are included in Figure 6.**

3. What are the K_d 's of the 2 reagents (or apparent K_d 's). No characterization of the binding constants are given. Better characterization of this would be beneficial.

Response: The sCore2 lectin is typically used at a working concentration of 1-5 $\mu\text{g}/\text{mL}$. This is sufficient to observe >10-fold differences in binding assays with diverse cell types, compared to sialidase treated negative control. For example, in the spectral flow live cell experiments, the reagents was used at 1 $\mu\text{g}/\text{mL}$. Based on binding-dose studies (Fig. 5k), we estimate an apparent K_D of ~3.5 $\mu\text{g}/\text{mL}$.

At the current time, we are unable to measure precise binding affinities using BLI or SPR since our protein is dimeric, and the carbohydrate scaffolds we use are also multivalent on a PAA (polyacrylamide) backbone. There is also no established/simple method to immobilize the glycan-PAA backbone on a biosensor. Monovalent receptors and analytes are needed for affinity measurement. Based on our prior SPR work, however, the on/off-rate of such measurements is typically high¹⁵. Due to the difficulty in measuring affinities, we believe that it is more useful to focus on the working concentrations needed to obtain new biological insight. In general, the measurements of protein-carbohydrate K_d is a non-trivial undertaking, and the findings may only marginally improve the interpretation of results presented in this manuscript.

4. On a practical note, some mention of how easy, or hard, it is to make these reagents would be good. What is the average yield per x of cells? Is there a possibility of bacterial production or yeast production of these in the future? This is often a limitation for reagents to be useful. Although bacterial and yeast production are mentioned in terms of the mutant libraries, a more pertinent point would be the potential for making these reagents. In addition, how stable are these reagents? This is often an issue with mammalian lectins and is one of the reasons that the field has focused on bacterial lectins and glycosidases for engineering.

Response: Typically, we purify ~100 μg of the lectin fusion-protein after transfecting a single 150mm HEK cell petri dish, and this is sufficient for a large number of assays. In typical assays, we use 1-

5µg/mL of the lectin for single-cell binding studies. **These numbers are provided in the revised Methods and Figure Legends.**

We agree that large-scale production is desirable. To address this we attempted bacterial expression by including our lectins as maltose-binding protein (MBP)-fusion proteins. Though the protein was expressed, it was unstable following MBP removal. This is consistent with previous studies that suggest challenges with expression of mammalian glycosyltransferases in bacteria ². Currently, we are scaling up these lectins using standard yeast (*S. cerevisiae*) in anticipation that this will be followed by protein expression in yeast och1 mutant cells in the near future. The latter cells would bear glycan binding proteins with shorter, human-like mannosylated glycans.

As far as stability is concerned, we have stored functional reagent at 4°C for >6 months without loss of activity. We have also aliquoted the reagents at 0.5-1mg/mL in buffer supplemented with 0.1% BSA for long-term storage at -80C. This is also stable. We have not yet attempted lyophilization as this is a third potential strategy.

5. Although the staining data is interesting, there is no discussion of whether this matches expectations for sCore2. Would it be possible to compare this to GCNT1 staining on at least a few tissues to see whether this would line up (which based on arguments made would be somewhat expected)?

Response: As suggested, we applied anti-GCNT1 polyclonal sheep antibody (Product No: AF7248, R&D systems) along with sCore2 in selected tissue: **i)** two breast cancer tissue and **ii)** one spleen tissue (**see new Supplemental Figure S10**). A strong correlation was observed in one of two breast tissue and partial overlap was noted for spleen. In general, sCore2 bound specific cells, while anti-GCNT1 staining was broader. The distinct patterns suggest that while GCNT1 may be necessary for sCore2 binding, it is not alone sufficient as the binding epitope of sCore2 is more complex.

Minor issues:

1. On p. 6 diCBM40 is described as having "weaker sialic acid dependence". The choice of words is inaccurate. The data clearly shows that the lectin is dependent on sialic acid. It seems more clear that there is lower binding affinity as might be expected for a carbohydrate binding module.

Response: We rephrased this sentence as suggested.

2. On p. 7 "these lectins did not also bind additional sialoglycans: should read: these lectins also did not bind.

Response: We rephrased this part as suggested.

References:

- 1 Zhang, C., Kelkar, A. & Neelamegham, S. von Willebrand factor self-association is regulated by the shear-dependent unfolding of the A2 domain. *Blood Adv* **3**, 957-968 (2019). <https://doi.org:10.1182/bloodadvances.2018030122>
- 2 Moremen, K. W. *et al.* Expression system for structural and functional studies of human glycosylation enzymes. *Nat Chem Biol* **14**, 156-162 (2018). <https://doi.org:10.1038/nchembio.2539>
- 3 Warkentin, R. & Kwan, D. H. Resources and Methods for Engineering "Designer" Glycan-Binding Proteins. *Molecules* **26** (2021). <https://doi.org:10.3390/molecules26020380>

- 4 Yang, L. *et al.* Sialic acid binding polypeptide. USA patent US Patent No. 11,434,479; Japanese Patent No. 7270141; Australian Patent No. 2018258251 (2002).
- 5 Liang, S. *et al.* Mutant glycosidases for labeling sialoglycans with high specificity and affinity. *Nat Commun* **16**, 1427 (2025). <https://doi.org/10.1038/s41467-025-56629-9>
- 6 Eshhar, Z., Waks, T., Gross, G. & Schindler, D. G. Specific activation and targeting of cytotoxic lymphocytes through chimeric single chains consisting of antibody-binding domains and the gamma or zeta subunits of the immunoglobulin and T-cell receptors. *Proc Natl Acad Sci U S A* **90**, 720-724 (1993). <https://doi.org/10.1073/pnas.90.2.720>
- 7 Beerli, R. R. *et al.* Isolation of human monoclonal antibodies by mammalian cell display. *Proc Natl Acad Sci U S A* **105**, 14336-14341 (2008). <https://doi.org/10.1073/pnas.0805942105>
- 8 Zhou, C., Jacobsen, F. W., Cai, L., Chen, Q. & Shen, W. D. Development of a novel mammalian cell surface antibody display platform. *MAbs* **2**, 508-518 (2010). <https://doi.org/10.4161/mabs.2.5.12970>
- 9 Javanmardi, K. *et al.* Rapid characterization of spike variants via mammalian cell surface display. *Mol Cell* **81**, 5099-5111 e5098 (2021). <https://doi.org/10.1016/j.molcel.2021.11.024>
- 10 Soga, K. *et al.* Mammalian Cell Surface Display as a Novel Method for Developing Engineered Lectins with Novel Characteristics. *Biomolecules* **5**, 1540-1562 (2015). <https://doi.org/10.3390/biom5031540>
- 11 Crook, Z. R. *et al.* Mammalian display screening of diverse cysteine-dense peptides for difficult to drug targets. *Nat Commun* **8**, 2244 (2017). <https://doi.org/10.1038/s41467-017-02098-8>
- 12 Gupta, R., Matta, K. L. & Neelamegham, S. A systematic analysis of acceptor specificity and reaction kinetics of five human alpha(2,3)sialyltransferases: Product inhibition studies illustrate reaction mechanism for ST3Gal-I. *Biochem Biophys Res Commun* **469**, 606-612 (2016). <https://doi.org/10.1016/j.bbrc.2015.11.130>
- 13 Kitagawa, H. & Paulson, J. C. Cloning of a novel alpha 2,3-sialyltransferase that sialylates glycoprotein and glycolipid carbohydrate groups. *J Biol Chem* **269**, 1394-1401 (1994).
- 14 Mondal, N. *et al.* ST3Gal-4 is the primary sialyltransferase regulating the synthesis of E-, P-, and L-selectin ligands on human myeloid leukocytes. *Blood* **125**, 687-696 (2015). <https://doi.org/10.1182/blood-2014-07-588590>
- 15 Beauharnois, M. E. *et al.* Affinity and kinetics of sialyl Lewis-X and core-2 based oligosaccharides binding to L- and P-selectin. *Biochemistry* **44**, 9507-9519 (2005). <https://doi.org/10.1021/bi0507130>